# Accurate auto-labeling of chest X-ray images based on quantitative similarity to an explainable AI model

Doyun Kim[1,2], Joowon Chung [1,2], Jongmun Choi[1], Marc D. Succi[1], John Conklin[1],
Maria Gabriela Figueiro Longo[1], Jeanne B. Ackman [1], Brent P. Little[1], Milena Petranovic[1],
Mannudeep K. Kalra [1], Michael H. Lev [1] & Synho Do [1✉]

The inability to accurately, efficiently label large, open-access medical imaging datasets limits the widespread implementation of artificial intelligence models in healthcare. There have been few attempts, however, to automate the annotation of such public databases; one approach, for example, focused on labor-intensive, manual labeling of subsets of these datasets to be used to train new models. In this study, we describe a method for standardized, automated labeling based on similarity to a previously validated, explainable AI (xAI) model-derived-atlas, for which the user can specify a quantitative threshold for a desired level of accuracy (the probability-of-similarity, pSim metric). We show that our xAI model, by calculating the pSim values for each clinical output label based on comparison to its training-set derived reference atlas, can automatically label the external datasets to a user-selected, high level of accuracy, equaling or exceeding that of human experts. We additionally show that, by fine-tuning the original model using the automatically labelled exams for retraining, performance can be preserved or improved, resulting in a highly accurate, more generalized model.

[1] Department of Radiology, Massachusetts General Brigham and Harvard Medical School, Boston, MA, USA. [2]These authors contributed equally: Doyun Kim, Joowon Chung. ✉email: sdo@mgh.harvard.edu

The implementation of medical artificial intelligence (AI) into clinical practice in general, and radiology practice in particular, has in large part been limited by the time, cost, and expertise required to accurately label very large imaging datasets, which can serve as platinum level ground truth for training clinically relevant AI models. The ability to automatically and efficiently annotate large external datasets, to a user-selected level-of-accuracy, may therefore be of considerable value in developing impactful, important, medical AI models that bring added value to, and are widely accepted by, the healthcare community. Such an approach not only has the potential to benefit retraining to improve the accuracy of existing AI models, but —through using explainable, model-derived atlas-based methodology[1]—may help to standardize labeling of open-source datasets[2–5], for which the provided labels can be noisy, inaccurate, or absent. Such standardization may, in turn, reduce the number of datapoints required for accurate model building, facilitating, training, and retraining from initial small but well annotated datasets[1,6].

In this study, we develop and demonstrate a method for standardized, automated labeling based on similarity to a previously validated explainable AI (xAI) model, using a model-derived atlas-based approach for which the user can specify a quantitative threshold for a desired level of accuracy (the probability-of-similarity, or pSim metric). The pSim values range from a "baseline" likelihood of similarity (pSim = 0, least selective) to a "maximal" likelihood of similarity (pSim = 1, most selective); pSim is computed by comparison between test-set derived image features and image features retrieved from the model's reference atlas (i.e., library). This model-derived atlas is constructed during model building (Fig. 1a) from the training set cases (Fig. 1a, b). The calculated pSim value reflects the harmonic mean between two model-related parameters, the "patch similarity" and the "confidence" (Methods, Fig. 1b, c).

Specifically, we applied our existing AI model for detection of five different chest X-ray (CXR) imaging labels (cardiomegaly, pleural effusion, pulmonary edema, pneumonia, and atelectasis), to three large open-source datasets—CheXpert[2], MIMIC[3], and NIH[4]—and compared the resulting labels to those of seven human expert radiologists. Of note, there is an inverse relationship between the selected pSim threshold values and the number of cases identified (i.e., captured) by the model from the external dataset; in other words, the higher the threshold for likelihood of similarity, the fewer cases that will be identified from the external database as similar to the model labeled cases.

We showed that our xAI model, by calculating the pSim values for each clinical output label based on comparison to the model's training-set derived reference atlas, could automatically label the external datasets to a user-selected, arbitrarily high level of accuracy, equaling or exceeding that of human experts. Moreover, we additionally showed that, by fine-tuning the original model using the automatically labeled exams for retraining, performance could be preserved or improved, resulting in a highly accurate, more generalized model. Although the pSim threshold values required to achieve maximal similarity vary by clinical output label, once those values are identified—based on comparison of model labels to a relatively small subset of expert-annotated ground truth labels—they can then be applied to the remaining external dataset, to identify exams likely to be positive for that clinical output label at a predetermined, high confidence level of accuracy; the resulting labels can then be applied for fine-tuning or retraining of the original model.

## Results
### System design
We developed an xAI model for detection of the following five different labels on posterior–anterior (PA) projection CXRs: cardiomegaly, pleural effusion, pulmonary edema, pneumonia, and atelectasis (see Methods). As per previous reports, our model featured atlas creation and prediction-basis calculation modules for explainability (Fig. 1)[1]. The prediction basis was used to calculate a patch similarity value (a probability between 0 and 1). Our model also included a confidence probability calculation module (Fig. 1a and b). The harmonic mean between the patch similarity and confidence model outputs were used to calculate a quantitative probability-of-similarity (pSim) value, between 0 and 1, for each clinical output label studied (Fig. 1c).

### xAI model development
CXR examinations performed at our institution from February 2015 through February 2019 were identified from our RIS (Radiology Information System) and PACS (Picture Archiving and Communication System), resulting in a dataset of 440,852 studies. Examinations were excluded if there was no associated radiology report, no view position information (e.g., anteroposterior projection, portable, etc.), or no essential patient identifiers (including but not limited to medical record number, age, or gender). A total of 400,886 CXR images from 267,180 examinations, representing 117,195 patients, together with their corresponding radiology reports, were collected retrospectively (Supplementary Fig. 1). Using a rule-based Natural Language Processing (NLP) model (Supplementary Table 1), we automatically extracted 20 pathological labels from the radiology reports, which were assigned one of the following three labels: positive, negative, or ignore. After automated NLP data mining and clean-up, we archived 151,700 anteroposterior CXR views from 49,096 patients (58% male, mean age $62 \pm 18$ years) and 90,023 posteroanterior (PA) CXR views from 69,404 patients (50% male, mean age $57 \pm 19$ years). We randomly selected 1000 images for each view position as a test set; the remaining examinations, from non-overlapping patients, were separated into training and validation sets (Supplementary Fig. 1). The labels for the training and validation sets were determined exclusively from the automated NLP assignments, whereas those for the test set were determined by consensus of three U.S. board-certified radiologists at our institution (further details provided in Supplementary Table 1), using the "Mark-it" tool (https://markit.mgh.harvard.edu, MA, USA) for annotation[7]. Our xAI model was trained by supervised learning with a total training dataset of 138,686 CXRs and achieved a mean Area Under the Receiver Operating Characteristic (AUROC) curve[8] of $0.95 + 0.02$ for detection of the five clinical output labels (Supplementary Table 2) in our initial, independent test set (Methods).

### Auto-labeling model performance applied to three open-source datasets
We applied our xAI CXR auto-labeling model to the available PA CXR images from three large open-source datasets: CheXpert ($n = 29,420$ PA CXR's), MIMIC ($n = 71,223$), and NIH ($n = 67,310$)[2–4]. To assess labeling accuracy, we randomly selected a subset of "positive" and "negative" cases as determined by the model for each of the five labels, distributed equally in each of ten pSim value ranges (0–0.1, 0.1–0.2, 0.2–0.3, ..., 0.9–1.0), for expert review (Figs. 2–4). Ground truth (GT) was defined as the majority consensus of seven expert sub-specialist radiologists (three with 12–25 years' experience in thoracic radiology and four with 1–6 years' experience in emergency radiology); GT and individual ratings of each reader, for each clinical output label (cardiomegaly, pleural effusion, pulmonary edema, pneumonia, and atelectasis), in each of the pSim value ranges, are shown in Figs. 2–4a (upper left). In Figs. 2–4b (upper right), we graph the relationship between the pSim value applied for the model's auto-labeling (x-axis) and both the (i) positive predictive value (PPV) and negative predictive

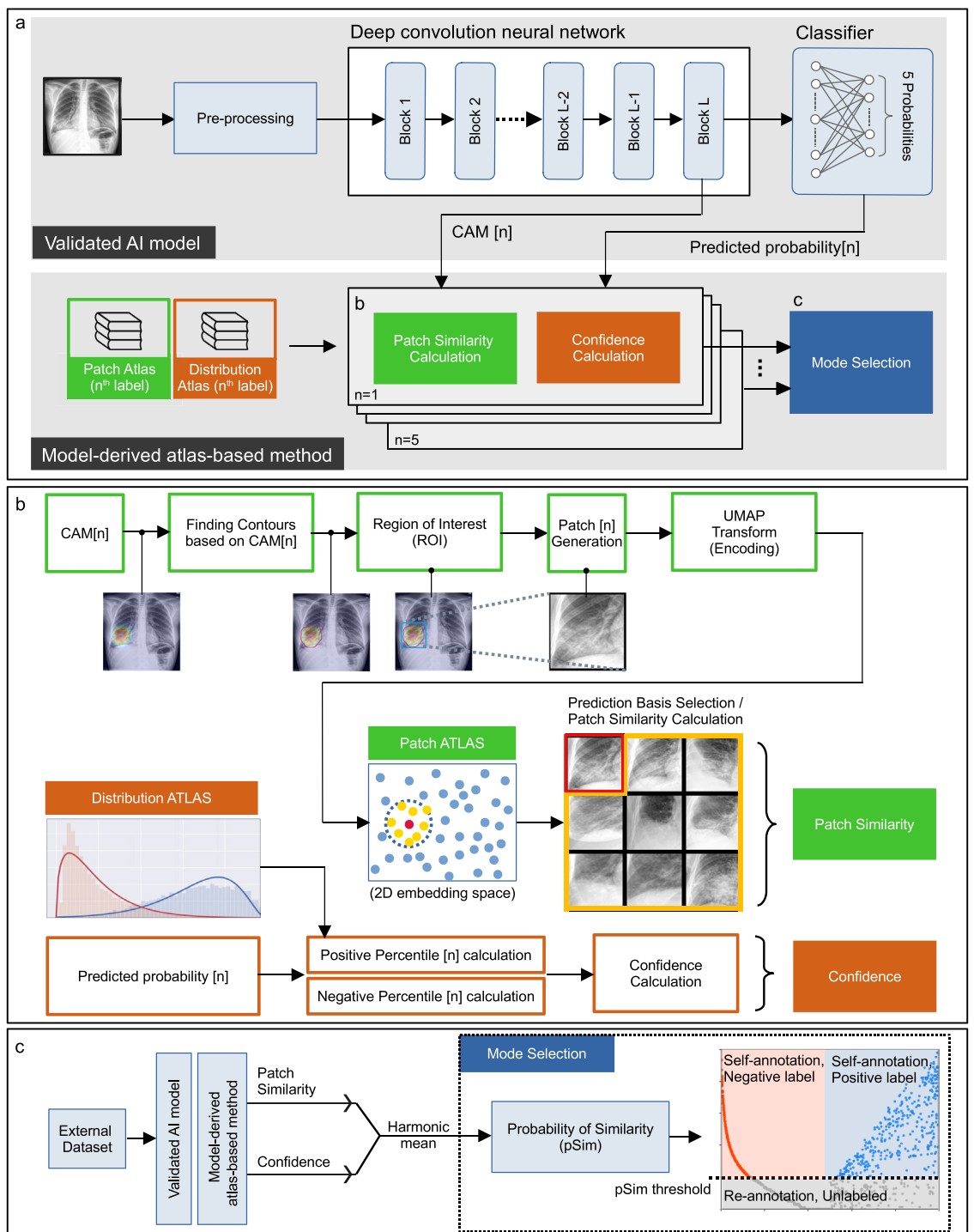

**Fig. 1 System overview.** Standardized, automated labeling method, based on similarity to a previously validated five-label chest X-ray (CXR) detection explainable AI (xAI) model, using an xAI model-derived-atlas based approach. **a** Our quantitative model-derived atlas-based explainable AI system calculates a probability-of-similarity (pSim) value for automated labeling, based on the harmonic mean between the patch similarity and the confidence. The resulting pSim metric can be applied to a "mode selection" algorithm, to either label the external input images to a selected threshold-of-confidence, or alert the user that the pSim value falls below this selected threshold. **b** The model-derived atlas-based method calculates patch similarity and confidence, based on class activation mapping (CAM)[38,39] and predicted probability from the model, for each clinical output label. **c** The harmonic mean between the patch similarity and confidence is then used to calculate a pSim for each clinical output label in mode selection.

value (NPV) of the model's ratings, versus ground truth; and the (ii) model's true positive capture rate (TPCR) and true negative capture rate, defined respectively as the total true positive (by GT) divided by the total positive (by GT), and the total true negative (by GT) divided by the total negative (by GT). In

Figs. 2–4c (lower left) and Figs. 2–4d (lower right), respectively, the number of false positive (by GT) and false negative (by GT) cases rated by the model at each pSim threshold value (x-axis), are shown, stratified by datasets (i.e., CheXpert, MIMIC, or NIH), with the optimal, lowest pSim threshold achieving 100% PPV or NPV,

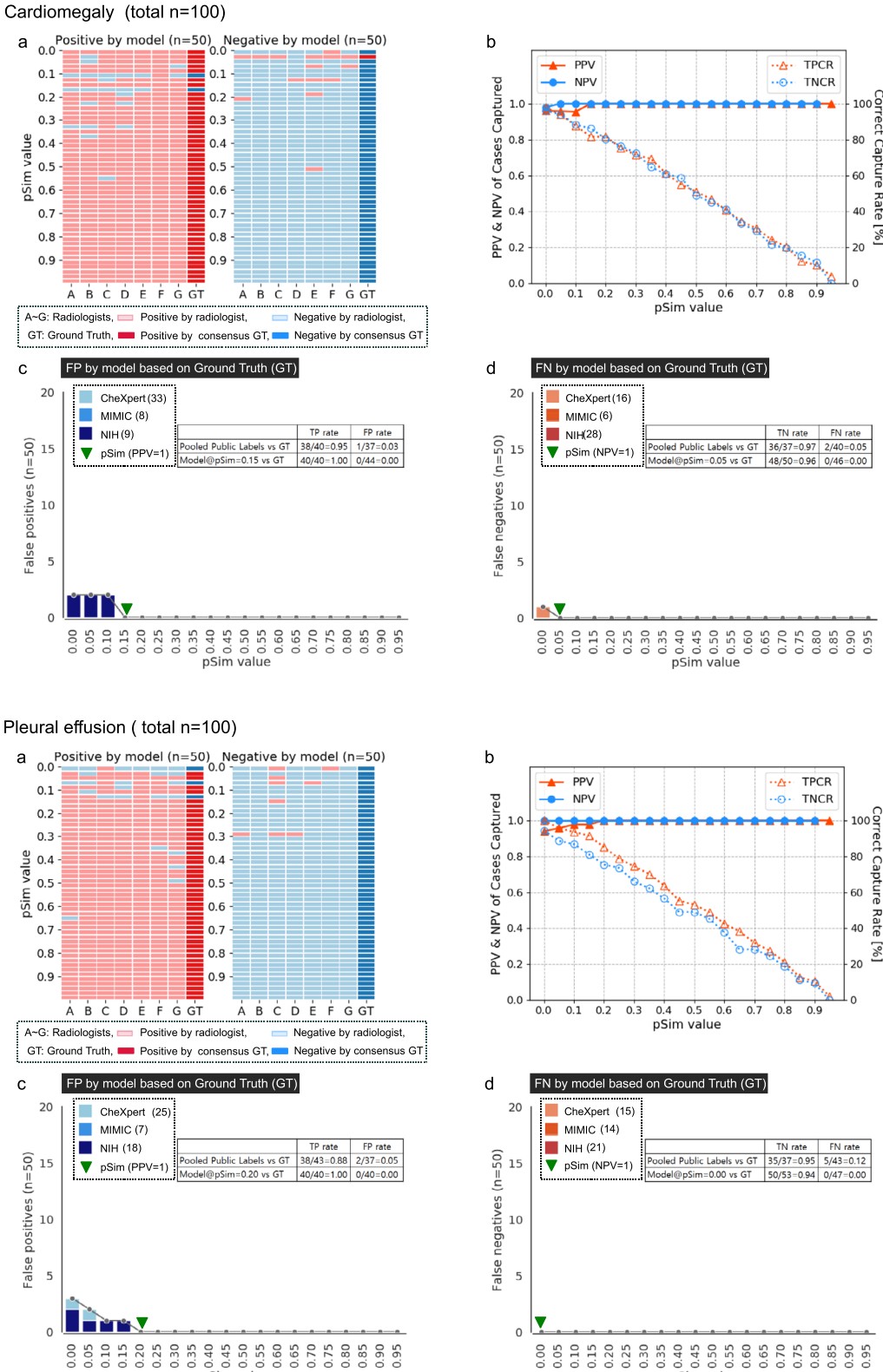

indicated. Of note, the lowest possible pSim threshold required for 100% PPV or NPV, corresponds to the maximal "correct capture rate", as shown in Figs. 2–4b.

Also, as shown in the text boxes in Figs. 2–4c, d, as well as in Fig. 5, model accuracy compared favorably to that of the available pooled public labels of the external, open-source datasets. Figure 5 additionally shows that the automated-labeling model's AUROC

performance, compared favorably to that of the individual expert radiologists, for each clinical output label, at both the pSim = 0 baseline value labeling threshold and the optimal pSim value labeling threshold (i.e., the lowest pSim value achieving 100% accuracy, as per Figs. 2–4c, d).

Sample auto-labeled CXR images that had complete agreement between all seven expert radiologists and the xAI model, positive

**Fig. 2 Automated-labeling model performance applied to three open-source CXR datasets, compared to consensus ground truth of seven expert radiologists, for the cardiomegaly & pleural effusion image labels.** We applied our xAI CXR auto-labeling model to three large open-source datasets: CheXpert, MIMIC, and NIH. For two of the five clinical output labels (cardiomegaly & pleural effusion), we randomly selected a subset of "positive" and "negative" cases as determined by the model, distributed equally in each of ten pSim value ranges (0–0.1, 0.1–0.2, 0.2–0.3, ..., 0.9–1.0), for expert review. In **a**, the positive (light red) and negative (light blue) ratings for each of the seven individual readers (columns A–G) are displayed graphically, with the consensus ground truth (GT, determined by majority) shown in the last column (bold red or bold blue). In **b**, the positive predictive values (PPV = [true positive by GT]/[total positive by model], solid red triangles, y-axis left) and negative predictive values (NPV = [true negative by GT]/[total negative by model], solid blue circles, y-axis left), of the model's ratings, are graphed versus the pSim threshold value that was applied by the model (x-axis). Also displayed in **b** (y-axis right) are the model's true positive capture rate (TPCR, dotted red triangles) and true negative capture rate (TNCR, dotted blue circles), defined respectively as TPCR = [true positive (TP) by GT]/[total positive by GT (number bold red from **a**)] and TNCR = [true negative (TN) by GT]/[total negative by GT (number bold blue from **a**)]. In **c** (lower left) and **d** (lower right), respectively, the number of false positive (FP by GT) and false negative (FN by GT) cases rated by the model at each pSim threshold value (x-axis), are shown stratified by dataset (CheXpert, MIMIC, or NIH; total number cases positive or negative by the model in parentheses), with the optimal, lowest pSim threshold achieving 100% PPV or NPV, as indicated (bold green triangles).

for each of the five clinical output labels studied, are shown in Supplementary Fig. 2. The pSim threshold values applied by the model for each image and the number/percent of PA CXR examinations with total agreement for each label, are also shown. Of note, there were only 14 positive examinations identified by the model as pneumonia that had full agreement with each reader, of 50 total examinations labeled as positive for pneumonia (28%). The percent positive labels with complete agreement for the other four labels, as shown in the figure, were cardiomegaly 78% (39/50), pleural effusion 78% (39/50), pulmonary edema 43% (17/40), and atelectasis 46% (23/50).

In Supplementary Table 3, we applied our automated-labeling model to the three complete public, open source CXR datasets: CheXpert ($n = 29,420$), MIMIC ($n = 71,223$), and NIH ($n = 67,310$); in order to demonstrate the magnitude of the number of cases captured, at the optimized pSim threshold value for maximal accuracy for each clinical output label (PPV, NPV = 1; as per Figs. 2–4). Pooling the model's labels for the three full public datasets (Supplementary Table 3, C) resulted in a capture rate of 80% for cardiomegaly (134,076/167,953), 68% for pleural effusion (114,230/167,953), 27% for pulmonary edema (45,660/167,953), 20% for pneumonia (33,308/167,953), and 28% for atelectasis (47,436/167,953). It is noteworthy that the model's mean CXR "capture rates" for the pooled results from the three public datasets, closely corresponded to those shown in the graphs of Figs. 2–4b, for the randomly selected subset of examinations ($n = 90$–100) labeled by both the model and the expert radiologists.

**Summary comparison of labeling efficiency, confidence metrics for the five auto-labeled clinical output labels**. For each of the five auto-labeled clinical output labels (Fig. 6), we compared: (i) the percent of positively auto-labeled CXR's captured from the three pooled, full public datasets (from Supplementary Table 3); (ii) the percent of cases with complete agreement between the model and all seven expert readers (from Supplementary Fig. 2); (iii) the lowest pSim value such that PPV = 1 (graphed as "1-pSim@PPV1"; from Figs. 2–4c), and (iv) the lowest pSim value such that NPV = 1 (graphed as "1-pSim@NPV1"; from Figs. 2–4d). Clinical output labels with higher values of these parameters (e.g., cardiomegaly, pleural effusion) corresponded to greater model auto-labeling efficiency and confidence; Clinical output labels with lower values (e.g., pulmonary edema, pneumonia) corresponded to lesser model auto-labeling efficiency and confidence. Of note, for atelectasis, "1-pSim@PPV1" was higher than "1-pSim@NPV1", indicating greater confidence that the model is correct in "ruling-in" this label (i.e., correctly auto-labeling true-positives) than in "ruling-out" this label (i.e., correctly auto-labeling true-negatives). This relationship was

reversed for the other four labels (e.g., greater confidence that the model can correctly "rule-out" than "rule-in" pneumonia or pulmonary edema).

The pairwise kappa statistics estimating inter-observer variability among the seven expert radiologists are shown in Fig. 7, for each of the five auto-labeled clinical output labels. The ranges for these values are as follows: cardiomegaly 0.82–0.92, pleural effusion 0.78–0.94, pulmonary edema 0.57–0.86, pneumonia 0.38–0.80, and atelectasis 0.47–0.78. The distribution of these ranges correlates well with the model's per clinical output label auto-labeling efficiency and confidence metrics, shown in Fig. 6, with cardiomegaly and pleural effusion showing the most inter-rater agreement, and pneumonia, pulmonary edema, and atelectasis showing the least.

In Fig. 8, we compare the model's auto-labeling performance using that pSim metric, to that of using either (1) patch similarity (based on CAM calculations, related to "focal" spatial localization) or (2) confidence probability (related to the "global" probability distribution of the final model output labels), alone. Our new analysis suggests that the use of a quantitative pSim threshold may have benefits over either patch similarity or confidence calculation alone, which is especially notable for those clinical diagnosis output labels—pneumonia and pulmonary edema—that have the lowest inter-rater agreement among experts (Fig. 7). These results impact the "explainability" of our model with regard to saliency maps. A recent paper concluded that saliency map techniques are highly variable, and that their use "in the high-risk domain of medical imaging warrants additional scrutiny"; the authors recommended "that detection or segmentation models be used if localization is the desired output of the network". A noteworthy feature of our approach, however, is its explainability based on quantitative pSim values (calculated from our model-derived-atlas), which as discussed, may have added value over saliency maps created using patch similarity or confidence calculations only[9].

We also studied the relationship between performance consistency, generalizability, dataset size, and architecture. Regarding architecture, there was excellent consistency between our current model and three additional, different model architectures, including ResNet-50[10], MobileNet v2[11], and MnasNet[12] (Supplementary Fig. 3). Our results similarly suggest consistent, robust generalizability regarding dataset size and heterogeneity (Table 1, Supplementary Tables 3 and 4).

To demonstrate the ability of our system to generalize to external datasets at a user designated level of performance, we fine-tuned our original model through iterative re-training using the auto-labeled CXR exams from the three public datasets (Table 1). The CXR exams selected for re-training ($n = 31,020$) had at least one positive label, a pSim value greater than or equal

**Table 1 Added value of fine-tuning with auto-labeled datasets for model generalizability and performance improvement.**

|  | Original Model (Ensemble) | Fine-tuned Model (Ensemble) |
|---|---|---|
| Cardiomegaly | 0.994 | 0.998 |
| Pleural Effusion | 0.998 | 0.998 |
| Pulmonary edema | 0.960 | 0.968 |
| Pneumonia | 0.908 | 0.930 |
| Atelectasis | 0.954 | 0.965 |
| Average Score | 0.963 | 0.972 |

To demonstrate the ability of our system to generalize to external datasets at a user designated level of performance, we fine-tuned our original model through iterative re-training using the auto-labeled CXR exams from the three public datasets. The CXR exams selected for re-training ($n = 31{,}020$) had at least one positive label, a pSim value greater than or equal to the optimal threshold for that label (as per Figs. 2–4c, 2–4d, and 5), and were excluded if they had been used previously as part of the test set. The ensemble performance of the original model, compared to the fine-tuned model, for each of the five model labels and their average, is shown (see also Supplementary Table 4 for the stratified performance of the six DenseNet-121 models that form the ensemble for each). The fine-tuned model utilized the same environments and hyper-parameters (e.g., learning rate = $10^{-8}$) as the original model. Performance for each of the model labels was preserved or improved on the more generalized, fine-tuned model.

to the optimal threshold for that label (as per Figs. 2–4c, 2–4d, and 5), and were excluded if they had been used previously as part of the test set. Our results comparing performance of the original model to that of the fine-tuned model (Table 1 and Supplementary Table 4), showed equal or improved accuracy of the fine-tuned model—trained using both local and more generalized data from the three public datasets—versus the original model, which was trained using local data only.

## Discussion
Accurate, efficient annotation of large medical imaging datasets is an important limitation in the training, and hence widespread implementation, of AI models in healthcare[13–22]. To date, however, there have been few attempts described in the literature to automate the labeling of such large, open-access databases[2–6]. One approach, for example, focused on developing new AI models using labor-intensive, manually annotated subsets of the external datasets, and applying these models to the remaining database[6]. The accuracy of such an approach can be limited not only by the: (1) baseline performance of the model, but also by (2) differences in the case mix and image quality of the external datasets. Moreover, as demonstrated by the results of our study, (3) it cannot be assumed that the labels provided with public databases are accurate or clean; for example, in some public datasets, such labels may have been generated from potentially noisy NLP derived annotation, without validation by an appropriate platinum level reference standard.

In this study, we demonstrate a method for standardized, automated labeling based on similarity to a previously validated xAI model, using a model-derived-atlas based approach, for which the user can specify a quantitative threshold for a desired level of accuracy, the pSim metric. Specifically, we applied our existing AI model for detection of five different CXR clinical output labels (i.e., cardiomegaly, pleural effusion, pulmonary edema, pneumonia, and atelectasis), to three large public open-source datasets (i.e., CheXpert, MIMIC, and NIH), and compared the resulting labels to those of seven human expert radiologists.

We showed that our xAI model, by calculating the pSim values for each label based on comparison to its retrieved training-set derived reference atlas, could automatically label a subset of the external data at a user-selected, arbitrarily high level of accuracy, equaling or exceeding that of the human experts (Fig. 5).

Moreover, we additionally showed that, by fine-tuning the original model using the automatically labeled exams for retraining, performance could be preserved or improved, resulting in a highly accurate, more generalized model.

The pSim value used for annotation reflects a trade-off between the accuracy of image labeling (i.e., the higher the pSim value, the more accurate the labels) and the efficiency of image labeling (i.e., the higher the pSim value, the fewer examinations that the model selects for annotation). To determine the pSim threshold for each output label such that PPV, NPV = 1, we randomly selected a subset of "positive" and "negative" exams from the three pooled open-source databases, distributed equally in each of ten pSim value ranges (0–0.1, 0.1–0.2, 0.2–0.3, …, 0.9–1.0) as per Figs. 2–4 (10 exams per pSim range for a total of 100). It is noteworthy that, using this approach for exam selection, we were able to achieve a very high level of labeling accuracy and model performance after fine-tuning, despite the relatively small number of cases presented for human expert review ($n = 100$).

To evaluate the efficiency of our automated-labeling approach, we applied our xAI model to the three full public datasets, and compared the five auto-labeled clinical output labels according to the following parameters: (i) the percent of positively auto-labeled CXR's from the three pooled public datasets (i.e., the capture rate), (ii) the percent of cases with complete agreement between the model and all seven expert readers, (iii) the lowest pSim value for annotation such that all positive cases captured are true positive (i.e., optimal pSim for PPV = 1), and (iv) the lowest pSim value for annotation such that all negative cases captured are true negative (i.e., optimal pSim for NPV = 1). We found a strong correlation between the magnitude of these parameters for each of the annotated clinical output labels, as shown in Fig. 6. It is noteworthy that the positive capture rates from the three pooled public datasets also strongly correlated with the capture rates graphed in Figs. 2–4b, for the subset of examinations ($n = 90$–100) labeled by both the model and the radiologist experts. Moreover, the parameter values reported for each clinical output label corresponded well with the kappa values for inter-observer variability shown in Fig. 7.

Together, our results suggest that the overall accuracy and efficiency of the auto-labeling model, applied to the full public datasets at the optimal pSim for each clinical output label, may be similar to the accuracy and efficiency of the model as applied to the subset of examinations annotated by the seven expert radiologists. These results also suggest greater auto-labeling efficiency, with higher confidence in label accuracy, for cardiomegaly and pleural effusion—two of the more objective findings in CXR interpretation—and lesser auto-labeling efficiency, with lower confidence in label accuracy, for pneumonia and pulmonary edema—two of the more subjective assessments in CXR interpretation. Indeed, the larger the quantity "$1$-pSim$_{optimal}$" for a given clinical output label (where $0 \leq \text{pSim} \leq 1$ and pSim$_{optimal}$ = the minimum pSim value such that PPV/NPV = 1), the more reliable and robust is the labeling for that clinical output label, based on similarity to the "remembered" reference atlas derived from the model's NLP training set.

An important feature that distinguishes our approach from that of other black-box classification models is explainability; the pSim metric provides feedback that the model is performing at a predetermined level of accuracy. Labeling external datasets using black-box classification methods is likely to be more labor-intensive than with our approach, because each distinct dataset (e.g., CheXpert, NIH, and MIMIC) may require a greater number of manual labels to ensure that sufficient representative exams have been sampled. Using pSim to estimate a quantitative probability-of-similarity, however, could provide greater user confidence that sufficient exams have been sampled for accurate

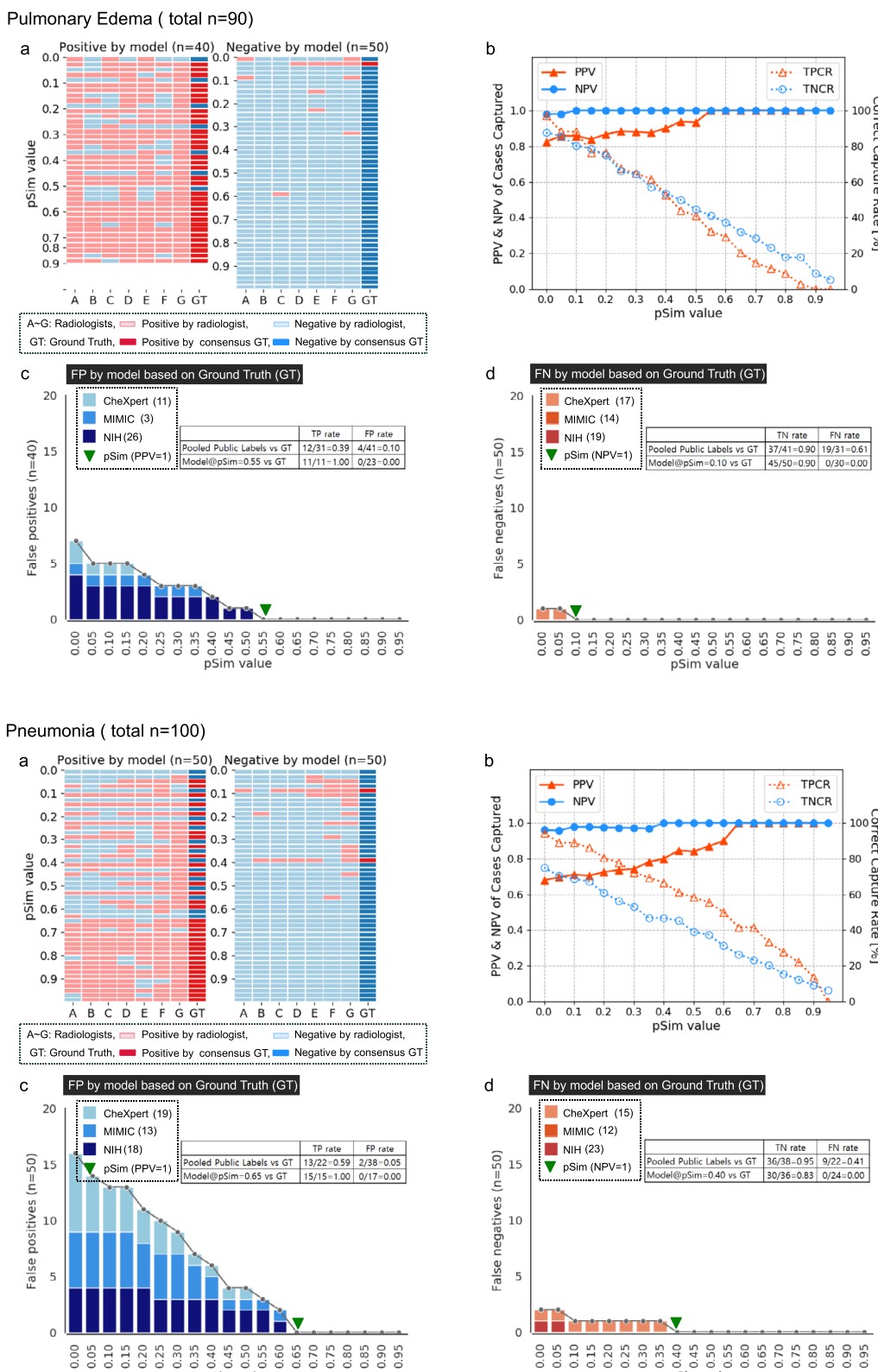

**Fig. 3 Automated-labeling model performance applied to three open-source CXR datasets, compared to consensus ground truth of seven expert radiologists, applied to the pulmonary edema and pneumonia labels.** Please refer to Fig. 2 for **a–d** captions.

model performance. In the future, such expert manual annotation might only need to be done once for any given platform at any given institution, facilitating automated continuous fine-tuning and retraining. Indeed, a recent paper found that "for a brain lesion segmentation model trained on a single institution's data,

performance was lower when applied at a second institution; however, the addition of a small amount (10%) of training data from the second institution allowed the model to achieve its full potential performance level at the second institution". Our approach has the potential to facilitate fine-tuning or retraining to

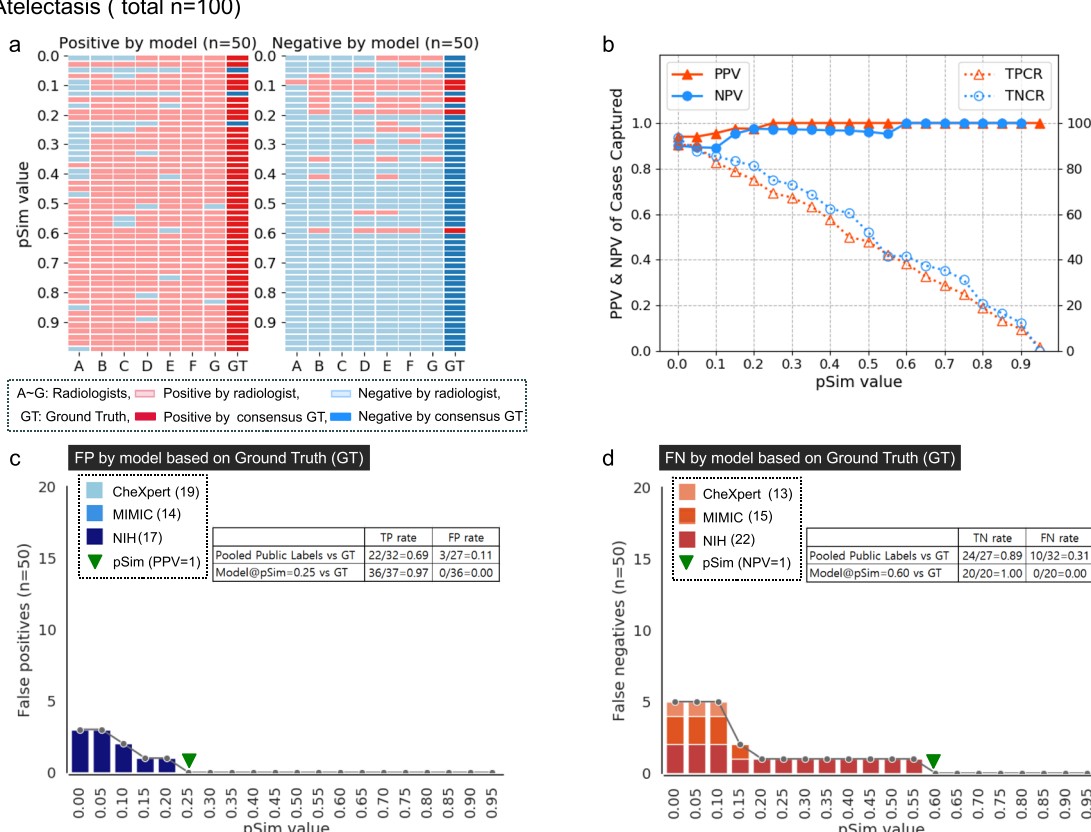

**Fig. 4 Automated-labeling model performance applied to three open-source CXR datasets, compared to consensus ground truth of seven expert radiologists, applied to the atelectasis label.** Please refer to Fig. 2 for **a–d** captions.

a similar or greater level of performance, using considerably less data than 10% of the initial training set[23].

Another noteworthy aspect of our approach relates to system deployment. We can apply the pSim value threshold to each class independently, selecting a low pSim value for high conspicuity clinical output label with high inter-rater agreement, and selecting a high pSim value for noisier, more subjective non-specific clinical output label with lower inter-rater agreement, the latter at the cost of generating fewer labeled examinations (i.e., lower capture rate). Employing pSim values helps quantify which clinical output labels of the AI model are most reliably annotated and which need to be improved, making it possible to measure system robustness. Deploying the xAI system is also HIPAA compliant, as no patient identifiable source data need be stored, since the mode selection (Fig. 1) uses only the encoded predicted probability distributions for categories and the compressed information from the UMAP transformation[24] for the atlas.

Other current approaches to auto-labeling have involved semi-supervised[6,25] and self-supervised[26–29] learning. Because these approaches assume low correlation between classes, however, their performance has not been validated for multi-label CXR classification models with high interclass correlation. Transfer learning and fine-tuning have also been attempted to improve performance when independently developed models are applied to external datasets[30–32], however, these methods are often impractical because different institutions are likely to use different definitions for similar categories, and capturing data with external labels based on even slightly different definitions can introduce considerable noise when such data is used for training or retraining new models. Our approach, however, allows for generation of standardized labels, with a user defined probability-of-similarity to that of established models. Our model-derived atlas-based approach, which simplifies the

computational issues by focusing on small patch regions with lower interclass and higher intraclass correlations, could achieve high accuracy and efficiency for auto-labeling three large public open source CXR datasets, similar to or exceeding that of human experts.

Our auto-labeling AI model reflects several characteristics of human intelligence[33] in general, and radiologist-mimicking behavior in particular. Specifically, our system is "smart", in that it can access its "memory" of examination clinical output labels present in the training set, and quantitatively estimate their similarity to clinical output labels in the new, external examination data. The "1-pSim_{optimal}" metric for each clinical output label provides a measure of the "intelligence" of the system for efficient accurate labeling, and its value (between 0 and 1) reflects the quality (i.e., ground-truth accuracy) of the NLP-derived dataset used for initial training. The model can also provide feedback to users through its explainability functionality, by displaying examples of the clinical output labels under consideration from its reference atlas together with their associated pSim value; this interaction offers the user an additional level of confidence that the model is doing what it's supposed to do. In this regard, our system can be viewed as an augmented intelligence tool to improve the accuracy and efficiency of medical imagers.

Indeed, one limitation of our model is that its labeling accuracy and efficiency is directly proportional to the quality of the initial training set. This may help explain why cardiomegaly and pleural effusion - two high-conspicuity clinical output labels routinely correctly described in the radiology reports identified by NLP for model training - have higher efficiency metrics (Figs. 2 and 6) than pulmonary edema and pneumonia (Fig. 3), which are more non-specific and variably assessed by different radiologists. This also may help explain why the 1-pSim_{optimal} values for NPV = 1 in Fig. 6 are higher than the 1-pSim_{optimal} values for PPV = 1, for

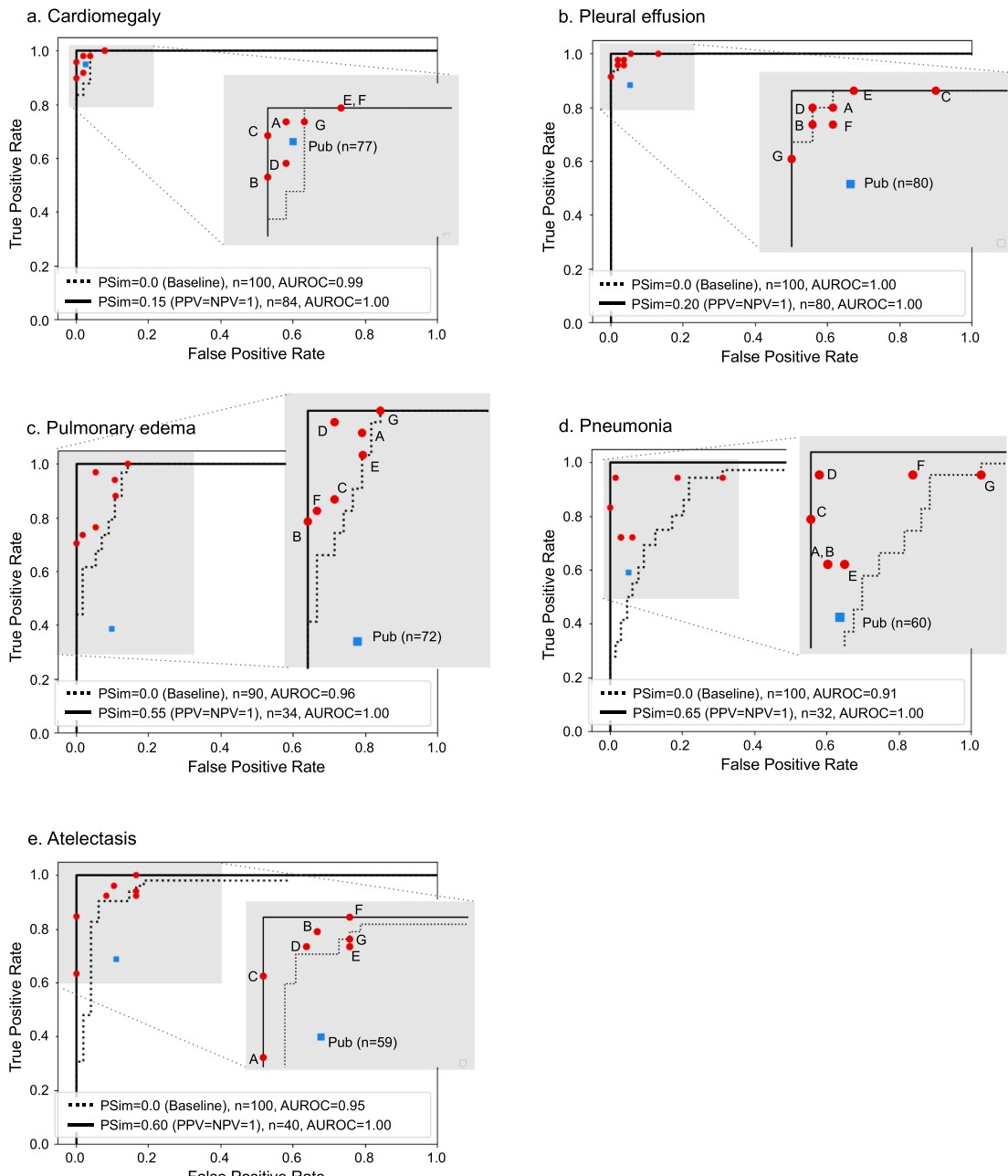

**Fig. 5 AUROC performance of automated-labeling model at two different pSim threshold values, compared to sensitivity, specificity of individual expert radiologists, and pooled public labels from three open-source CXR datasets.** AUROC performance of our xAI CXR auto-labeling model applied to the CheXpert, MIMIC, and NIH open-source datasets, is shown for each of the five labeled clinical output labels: **a** cardiomegaly, **b** pleural effusion, **c** pulmonary edema, **d** pneumonia, and **e** atelectasis. Comparison is to the performance of the individual expert radiologists (A–G, red circles), as well as to the performance of the pooled external annotations (blue squares, n = number available labeled external cases per clinical output label). ROC curves (y-axis sensitivity, x-axis 1-specificity) are shown for both the baseline pSim = 0 threshold (magnified box) and the optimal pSim threshold (i.e., the lowest pSim threshold achieving 100% accuracy, as per Figs. 2–4c and d).

all clinical output labels except atelectasis (Fig. 4), since atelectasis is a lower conspicuity, more non-specific clinical output label typically noted in CXR radiology reports only when it is present, but not mentioned when it is absent (i.e., the model learned from its NLP derived training set to have a higher level of certainty, and hence a higher $1\text{-pSim}_{optimal}$ value, when atelectasis is present, than when it is absent). Pulmonary edema and pneumonia, on the other hand, are typically described in CXR reports with a higher level of certainty when they are definitely absent (e.g., no evidence of pulmonary edema or pneumonia), than when they

are possibly present (e.g., cannot exclude pulmonary edema or pneumonia).

Moreover, because cardiomegaly and pleural effusion are focal, high-conspicuity regional imaging findings, they also demonstrate a higher TPCR performance with patch similarity than with confidence probability (Fig. 8). Similarly, for atelectasis, typically a more discrete, focal, regional CXR finding than pulmonary edema or pneumonia, both patch similarity and pSim (Fig. 8) show good TPCR performance relative to confidence probability. Conversely, for pulmonary edema, the only label for which TPCR

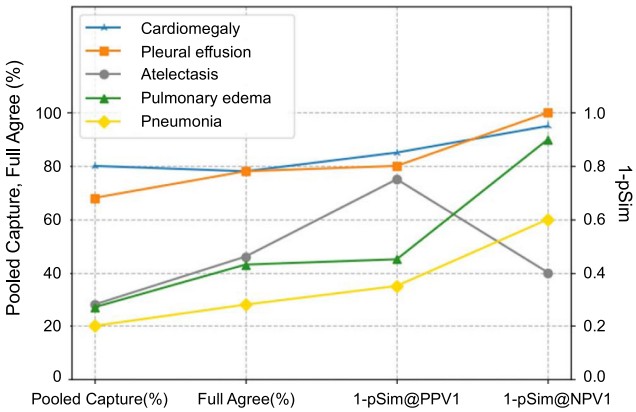

**Fig. 6 Comparison of labeling efficiency/confidence metrics for each of the 5 clinical output labels.** For each of the five auto-labeled clinical output labels– cardiomegaly (blue), pleural effusion (orange), atelectasis (gray), pulmonary edema (green), and pneumonia (yellow)—we compared: (i) the percent of positively auto-labeled CXR's "captured" from the three pooled, full public datasets (i.e., "Pooled Capture%", from Supplementary Table 3, C); (ii) the percent of cases with complete agreement between the model and all seven expert readers (i.e., "Full Agree%", from Supplementary Fig. 2); (iii) the lowest pSim value such that PPV = 1 (graphed as "1-pSim", from Figs. 2–4, c), and (iv) the lowest pSim value such that NPV = 1 (graphed as "1-pSim", from Figs. 2–4, d). clinical output labels with higher y-axis values (e.g., cardiomegaly, pleural effusion) correspond to those with greater model auto-labeling efficiency/confidence; clinical output labels with lower y-axis values (e.g., pneumonia, pulmonary edema) correspond to those with lesser model auto-labeling efficiency/confidence. Of note, in the graph for atelectasis, "1-pSim@PPV1" is higher than "1-pSim@NPV1", which can be interpreted as greater confidence that the model is correct in "ruling-in" the clinical output label (i.e., correctly auto-labeling true-positives) than in "ruling-out" the clinical output label (i.e., correctly auto-labeling true-negatives); this relationship is reversed for the other four clinical output labels (e.g., greater confidence that the model can correctly "rule-out" than "rule-in" pneumonia or pulmonary edema).

performance is better with confidence probability than with patch similarity (Fig. 8), this result is consistent with the fact that confidence probability is more sensitive for the detection of global, non-localized features, which are routinely associated with pulmonary edema findings on CXR (i.e., pulmonary edema is visualized diffusely throughout the bilateral lung fields).

It is noteworthy that the explanation for these differences in performance between confidence probability, patch similarity, and pSim for the five different labels (Fig. 8), corresponds so closely with the reader performance and reader variability shown in Figs. 2–5 and 7. This not only confirms our "common sense" clinical insight that cardiomegaly and pleural effusion (as well as atelectasis) are high conspicuity objective CXR findings, whereas pulmonary edema and pneumonia are more non-specific subjective assessments, but also underscores the explainability of our model (through assignment of appropriate pSim values for each label) in mirroring human performance, likely attributable to the radiologist based ground truth used for model training.

Another limitation of our model is that our proposed xAI system requires substantial computational resources and storage space to provide the prediction basis and to operate the mode selection module. Because the explainable modules have been designed to operate independently, however, we can differentially deploy the xAI system of adjusted capabilities according to the specification of a given server.

In summary, we have: (i) developed and demonstrated an explainable AI model for automated labeling of five different CXR

imaging clinical output labels, at a user selected quantitative level of confidence, based on similarity to the model-derived-atlas of an existing validated model, and (ii) showed that, by fine-tuning this existing model using the automatically labeled exams for retraining, performance could be preserved or improved, resulting in a highly accurate, more generalized model. It is noteworthy that these results were accomplished by human expert annotation of only 100 exams, selected from the three large independent datasets, representing an equal distribution of pSim threshold values from 0 to 1; this suggests that our approach based on quantitative similarity to an explainable AI model-derived-atlas may be able to provide highly accurate, fully automated labeling, regardless of the size of the open-source database being studied.

In conclusion, the ability to automatically, accurately, and efficiently annotate large medical imaging databases may be of considerable value in developing important, high-impact AI models that bring added value to, and are widely accepted by, the healthcare community. Our approach might not only help to improve the accuracy of existing AI models through fine-tuning and retraining, but also help to standardize labels of open-source datasets (for which the provided labels can be noisy, inaccurate, or absent) based on their quantitative similarity to those of existing, validated models. Use of the pSim metric for auto-labeling has the potential to reduce the amount of annotated data required for accurate model building, thereby reducing the need for labor-intensive manual labeling of very large datasets by human experts.

## Methods

This study was compliant with the Health Insurance Portability and Accountability Act and was approved by the Institutional Review Board of the Massachusetts General Hospital for retrospective analysis of clinically acquired data with a waiver of informed consent.

**Retrospective collection of the development and test datasets.** The development dataset contained CXR images acquired between February 2015 and February 2019. All DICOM (digital imaging and communications in medicine) images were de-identified before data analyses. To make a consistent dataset, we chose only examinations that had associated radiology reports, view position information (e.g., AP/PA projections, portable, etc.), and essential patient identifiers (including but not limited to medical record number, age, or gender). If an examination had multiple CXR images, only a single CXR image was included. We randomly selected 1000 images for each view position as a test set; the remaining examinations, from non-overlapping patients, were separated into training and validation sets (Supplementary Fig. 1).

**Labeling of the development and test datasets.** The labels for the training and validation sets were determined exclusively from the automated NLP assignments, whereas those for the test set were determined by consensus of three U.S. board-certified radiologists at our institution (further details provided in Supplementary Table 1) using the "Mark-it" tool (https://markit.mgh.harvard.edu, MA, USA) for annotation[7].

**Network training.** Densely Connected Convolutional Network (DenseNet-121)[34], which connects each layer to all other layers in a feed-forward method, was selected to develop the 20 pathologic labels detection and classification system. The pre-trained model, available from the official repository in Pytorch[35,36], was fine-tuned by supervised learning with our training dataset and the NLP's labels after the last fully connected layer with 1000 outputs and the first convolutional layer were replaced with 21 outputs (i.e., 20 pathologic labels and view position) and with inputs of 1 channel depth, respectively. The network topology was optimized using AdamW[37], where we used a batch size of 144, a learning rate of $1 \times 10^{-4}$, beta-1 of 0.9, beta-2 of 0.999, epsilon of $1 \times 10^{-8}$, and weight decay of $1 \times 10^{-5}$. In the training step, real-time data augmentation was performed by applying geometric transformations: rotation from −10 to 10, scaling to 110%, random crop to $512 \times 512$, random horizontal flip with 1% probability. All experiments were conducted on four GPUs of Tesla V100 SXM 32 GB [NVIDIA DGX, CA, USA], and all deep-learning models were implemented with Pytorch (v.1.2.0).

**Weighted loss function.** The Binary Cross-Entropy (BCE) loss function was weighted by the ratios of positive and negative samples for each class label ($\alpha_P^c$ and $\alpha_N^c$), for multi-label classification[4]. We considered two additional weights: the first weight had to reflect the ratio of the number of effective samples ($\alpha_e^c$, the maximal sum number between positive and negative labels among 20 clinical output labels divided

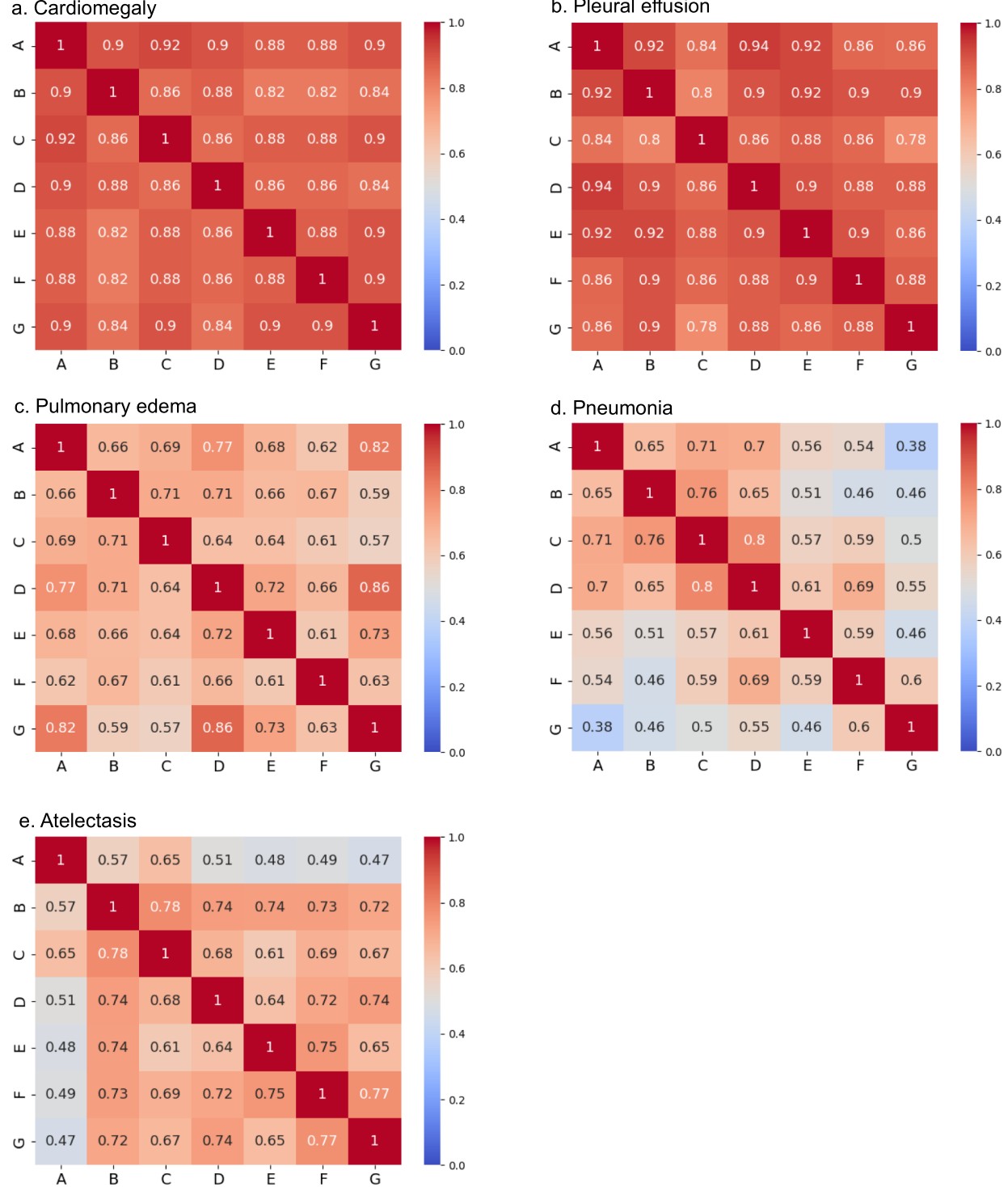

**Fig. 7 Pairwise kappa statistics between the seven expert radiologists, for each of the five clinical output labels.** For each of the five auto-labeled clinical output labels—**a** cardiomegaly, **b** pleural effusion, **c** pulmonary edema, **d** pneumonia, and **e** atelectasis—the pairwise kappa statistics estimating inter-observer variability are shown in the respective color-coded matrices[43].

by that of the c-th label) to train because of consideration of ignore labels for each clinical output label. When training the AI model, we experimentally found that using samples with the other view position as well as those with a targeted view position can improve the generalization performance of the model, so we added the second weight ($\alpha(v)$) in the loss function to relatively control the impact of samples with the target view position. The weighted BCE loss function is given by the Eq. (1):

$$L_{W-BCE}(\mathbf{x}, \mathbf{y}, \mathbf{t}, v) = -\alpha(v) \sum_{c=1}^{J} \alpha_s^c \{ \alpha_P^c t^c \ln y^c + \alpha_N^c (1 - t^c) \ln(1 - y^c) \} \quad (1)$$

where $\mathbf{x}$ denotes CXR images, the model's output is $\mathbf{y} = \{y^1, y^2, ..., y^J\}$ that indicates the predicted probability of $J$ classes, $v$ is a view position of the image, and $\mathbf{t} = \{t^1, t^2, ..., t^J\}$ means the labels of clinical output labels extracted by NLP. In addition, $\alpha_s^c$ is defined as $(|P^m| + |N^m|)/(|P^c| + |N^c|)$ in order to make fairness among classes with different numbers of effective samples which consider only "0" and "1", not "−1". Here, $|P^c|$ and $|N^c|$ are the total numbers of "1"s and "0"s in labels for c label, and $m$ means the class index having the maximum total number of both "1"s and "0"s ($m = \arg\max_c(|P^c| + |N^c|)$). We also define $\alpha_P^c = \frac{|P^c|+|N^c|}{|P^c|}$ and $\alpha_N^c = \frac{|P^c|+|N^c|}{|N^c|}$ for

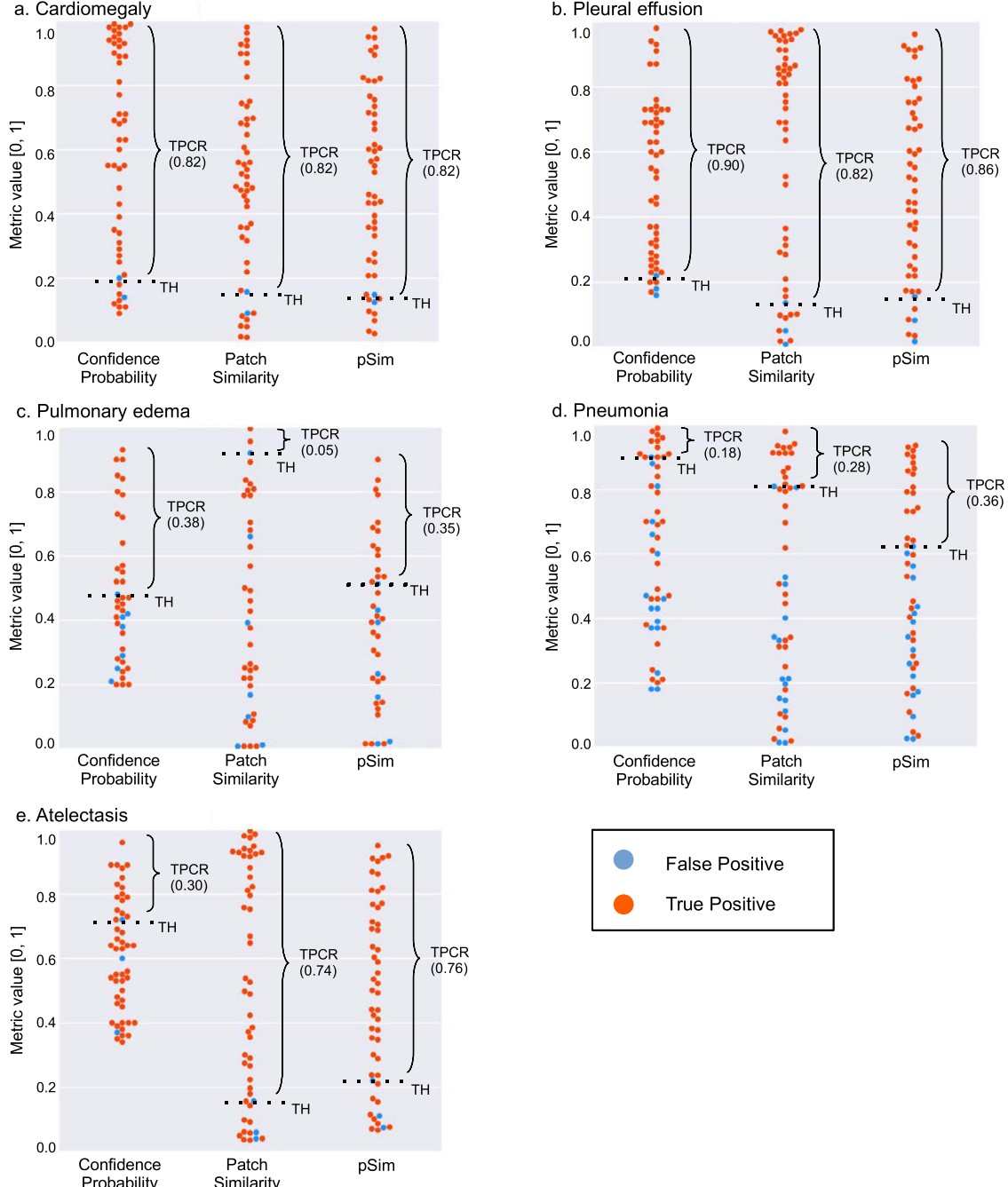

**Fig. 8 Performance comparison of Confidence Probability, Patch Similarity, and pSim in assigning true-positive model output labels for cardiomegaly, pleural effusion, pulmonary edema, pneumonia, and atelectasis.** We compared the true positive capture rate (TPCR) performance for each of the five clinical output labels, using confidence probability alone (reflecting the global probability distribution of the output labels), patch similarity alone (reflecting the focal spatial localization of the output labels), and pSim (reflecting the harmonic mean between the confidence probability and patch similarity, as per Fig. 1). These results are noteworthy in that the two model output labels that reflect high inter-rater agreement of imaging findings—**a** cardiomegaly and **b** pleural effusion, as per Fig. 7—show good agreement between the three confidence-level metrics, with high TPCR's for each. For the two output labels that show lower inter-rater agreement per Fig. 7—**c** pulmonary edema and **d** pneumonia—pSim performance significantly exceeds that of patch similarity for both, and that of confidence probability for pneumonia but not pulmonary edema. This difference is likely attributable to the fact that patch similarity is more sensitive for the detection of focal, regional imaging findings (e.g., as seen with the clinical diagnosis of pneumonia), whereas confidence probability is more sensitive for the detection of global findings (e.g., as seen with the clinical diagnosis of pulmonary edema). The results for **e** atelectasis, typically a more focal than global finding on CXR, may be similarly explained.

---

**Box 1 ▌ Mode selection for automated labeling method**

Input: predicted probability for c-class ($y^c$), $Confidence_P$, $Confidence_N$, and patch similarity
%[step-1] To divide into two groups by $y^c$ and $TH_{pos}$ : positive or negative candidates
If $y^c \geq TH_{pos}$: then
 %[step-2] To decide mode and annotation for the positive candidates
 % Probability-of-similarity, pSim
 pSim = 2 $Confidence_P$ pSimilarity / ($Confidence_P$ + pSimilarity)
 If pSim >= pSim threshold value (PPV, NPV = 1): then
 Mode = Self-annotation mode
 Label = 1%Positive label
 Else
 Mode = Re-annotation mode
 Label = -1%unlabeled
Else
 %[step-2] To decide mode and annotation for the negative candidates
 pSim = $Confidence_N$
 If pSim >= pSim threshold value (PPV, NPV = 1): then
 Mode = Self-annotation mode
 Label = 0 %Negative label
 Else
 Mode = Re-annotation mode
 Annotation = −1% unlabeled

---

solving the imbalance between positive and negative; $\alpha(v)$ is set to $\omega$ if $v$ is the targeted view, 1 for the others.

**Design overview for quantitative, explainable, model-derived atlas-based system.** Our automated dataset labeling, based on similarity to a validated CXR AI model, requires calculation of two quantitative atlas-based parameters, the "patch similarity" and "confidence" probabilities (values between 0 and 1), as per Fig. 1. For the "patch similarity" computation, a patch atlas is generated based on class activation mapping (CAM)[38,39]; for the "confidence" computation a distribution atlas is generated based on predicted probabilities (Fig. 1a, b). The harmonic mean between the patch similarity and confidence values are then used to calculate a pSim for each clinical output label (Fig. 1c).

**Predicted probabilities, model ensemble, and distribution atlas creation.** To improve the robustness of the entire system, an ensemble of six DenseNet-121 models is composed using unweighted averaging, such that the final probability is determined as an average of probabilities predicted by the six models[40]. Those six models are constructed by independently training with three weights (i.e., $\omega = 1.1$, 1.5, and 2.0 in $\alpha(v)$) for PA view, then selecting two models maximized by AUROC and accuracy, respectively. To create the Distribution-atlas, we do inference with the trained AI model on a full training dataset, to obtain two probability distributions of positive and negative samples for the training dataset. These probability distributions are saved as the Distribution-atlas for each clinical output label.

**Patch atlas creation based on CAM ensemble method.** To improve the localization performance of our class activation mapping, we developed an ensemble method as follows: by removing noise components of a single CAM, adding only significant components, and normalizing it in Eq. (2), the ensemble CAM was able to highlight sharply the overlapping regions among the single CAMs.

$$\mathbf{CAM_E^c} = Normalize\left(\sum_{s=1}^{S} \mathbf{CAM_s^c} \odot \mathbf{U_\tau}\right) \quad (2)$$

where $\mathbf{CAM_E^c}$ means the ensemble CAM matrix, $\mathbf{CAM_s^c}$ is a CAM matrix for the c-class generated from s-th single model, and S denotes the number of models. $\mathbf{U_\tau}$ denotes a matrix with the component of $u_{i,j} = u(\mathbf{CAM_s^c}(i,j) - \tau)$ to determine CAM values less than $\tau$ as noise components and to remove them. u is a unit step function, $\odot$ means the Hadamard product, and Normalize is a linear scale for converting into a standard range between 0 and 1.

To create the patch atlas, we search for main contours on a high-resolution CAM (512 × 512) generated from a CAM for each class, select a bounding box to include the outline, define it as the patch, and save it (one or two patches from a CAM are considered in this study). For each clinical output label, patches are saved as typical, representative patterns from only the CXR images with the AI model's predicted probability of being greater than or equal to 0.9. We train a cosine metric-based UMAP model using the patches for all clinical output labels[24]. The UMAP model transforms the patches into coordinates in two-dimensional embedding space, such that the smaller the Euclidean distance in this space, the higher the cosine similarity. For the automated labeling method, therefore, the patch atlas consists of coordinates for all patches in the

two-dimensional embedding space and the UMAP model (Fig. 1b). In addition, the patch atlas can be created using more advanced schemes[41,42].

**Patch similarity value calculation.** To calculate the patch similarity as shown in Fig. 1b, we need to extract the Prediction-basis ($\mathbf{\Psi_{pb}^c}$) for the c-th label by calculating Euclidean distance between the UMAP transformed coordinate of the input image and the Patch-atlas, and then by selecting K-basis with the minimum distance as Eq. (3):

$$\mathbf{\Psi_{pb}^c} = \left\{\mathbf{\Omega_{pb}^c}(1), ..., \mathbf{\Omega_{pb}^c}(K)\right\} \quad (3)$$

where $\mathbf{\Omega_{pb}^c}(k)$ denotes the patch with the k-th minimum Euclidean distance among the Patch-atlas, and the Euclidean distance is calculated by $\left\|f_{UMAP}^c(\mathbf{y_P^c}) - A_{P-UMAP}^c(i)\right\|_2$ for $i = 1, ..., n(A_{P-UMAP}^c)$. Moreover, $f_{UMAP}^c$ is the trained UMAP model for c-class, $\mathbf{y_P^c}$ is a 1024-dimensional patch vector calculated by an input image, $A_{P-UMAP}^c$ is the Patch-atlas, and $n(A_{P-UMAP}^c)$ is the size of the Patch-atlas. The patch similarity is proposed to enable the AI model to interpret the new patch based on the prediction-basis ($\mathbf{\Psi_{pb}^c}$), as a quantitative metric. The metric is calculated by a percentile of how close a patch of an input image is on a prediction-basis of K patches in the embedding space.

$$patch\ similarity = 1 - f_D^c\left(\frac{1}{K} \cdot \sum_{m=1}^{K}\left\|f_{UMAP}^c(\mathbf{y_P^c}) - f_{UMAP}^c(\mathbf{\Omega_{pb}^c}(m))\right\|_2\right) \quad (4)$$

where $f_D^c$ denotes a function calculating a percentile for the mean Euclidean distance of K-nearest patches for the input image, based on a distribution of the mean Euclidean distance for all patches of the Patch-atlas.

**Confidence value calculation.** As per Fig. 1b, we propose the confidence metric, based on the distribution atlas, as a measure of the trust level between the positive and negative predicted probabilities for a clinical output label. This quantitative metric is simply defined with Eqs. (5) and (6) for positive and negative predicted samples, as follows:

$$Confidence_P = \max\left(f_P^c(y^c) - (1 - f_N^c(y^c)), 0\right) \quad (5)$$

$$Confidence_N = \max\left((1 - f_N^c(y^c)) - f_P^c(y^c), 0\right) \quad (6)$$

Assuming that a predicted probability is $y^c$ for c-class, we calculate a percentile ($f_P^c(y^c)$) in the positive Distribution-atlas and a percentile ($1 - f_N^c(y^c)$) in the negative Distribution-atlas. Then, the difference between two percentiles is calculated as the confidence. Because the predictive ability of the xAI model for each clinical output label is related to the shape and degree of intersection of the two probability density curves (positive and negative) on the distribution-atlas, the confidence metric, as defined based on Eqs. (5) and (6), provides a quantitative measure analogous to a $p$ value between different statistical distributions. In other words, the higher the confidence value for a label, the higher the likelihood that the input image is mapping to the correct label, and the lower the likelihood of incorrect mapping. Moreover, this metric has the ability to quantify different levels of confidence according to different

distributions of clinical output label characteristics on the distribution atlas for each class of the model, even at the same predicted probabilities.

**pSim calculation, pSim threshold selection**. Our automated dataset labeling method calculates the pSim value using a harmonic mean between confidence and patch similarity (pSimilarity in Eq. (7)) for each input image.

$$pSim = 2 \cdot confidence \cdot pSimilarity / (confidence + pSimilarity) \qquad (7)$$

The pSim threshold for each clinical output label is chosen by the lowest pSim values that can achieve 100% PPV and NPV, as per Figs. 2–4.

Additional functionality of our model design includes a "mode selection" algorithm, which, using the selected pSim threshold value, can be used to either: (1) determine the image label (positive, negative, or unlabeled) within a given level-of-confidence if the pSim value for a class is greater than the selected threshold ("self-annotation mode"), or (2) alert the human user if the pSim falls below the selected threshold for level-of-confidence ("re-annotation mode"). Although the "re-annotation mode" was not applied to our current study, this has the potential to be of value in future applications and deployment of our model, as part of its explainability functionality (more details regarding pSim "mode selection" are provided in Methods Box 1).

**Statistical analyses**. To assess the statistical significance of the AUROC's, we calculated 95% CIs using a nonparametric bootstrap approach via the following process: first, 1000 cases were randomly sampled from the test dataset of 1000 cases with replacement, and the DCNN models were evaluated on the sampled test set. After running this process 2000 times, 95% CIs were obtained by using the interval between 2.5 and 97.5 percentiles from the distribution of AUROCs. The 95% CIs of percentage accuracy, sensitivity, and specificity of the models at the selected operating point were calculated using binomial proportion CIs.

**Selection of PA CXR's from three open-source datasets**. Although the external datasets contained both AP and PA views, our study was conducted with PA views only, for both consistency/convenience and to minimize potential confounding variables. Specifically, from the CheXpert v1 ($n = 223,414$) and NIH ($n = 112,120$) datasets, which contain PA labels in their metadata files, we collected 29,420 and 67,310 PA CXRs respectively. From the MIMIC v1 ($n = 369,188$) dataset, which did not have clear labels, we applied an internal model to distinguish between PA and AP projections, which returned 71,223 PA CXR's (specificity 0.999, sensitivity = 0.998).

**Reporting summary**. Further information on research design is available in the Nature Research Reporting Summary linked to this article.

## Data availability
The labels for the five categories applied to the three open datasets by the seven expert readers can be accessed at: https://github.com/MGH-LMIC/AutoLabels-PublicData-CXR-PA. The training, validation, and test datasets generated for this study are anonymized; the non-DICOM image format of this data may be available in 15 business days for research purposes from the corresponding author (sdo@mgh.harvard.edu) with an official request.

## Code availability
The codes for model development can be accessed at: https://github.com/MGH-LMIC/CXR-autolabeling.

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

## Acknowledgements

Our research provides a method to make practical use of open datasets. We appreciate CheXpert, MIMIC, and NIH, who have already put a lot of time and effort into sharing Chest X-ray images. We would also like to thank Thomas J. Schultz and Eric Michael L'Italien of the Enterprise Medical Imaging (EMI) team and Sehyo Yune, Myeongchan Kim, and Jan Sylwester Witowski at Massachusetts General Hospital's Radiology Department for their data curation assistance. And thanks to Nvidia and the Center for Clinical Data Science (CCDS) for making the DGX system available for our research.

## Author contributions

D.K., J.J., M.H.L., and S.D. initiated and designed the research. D.K., J.C., J.M.C., and S.D. curated data. M.D.S., J.C., M.G.F.L., J.B.A., B.P.L., M.P., and M.K.K. interpreted and annotated the data. D.K., J.J., M.H.L., and S.D. analyzed the data and results. D.K., J.J., J.B.A, M.P., B.P.L., M.H.L., and S.D. wrote the manuscript.

## Competing interests

M.H.L. is a consultant for GE Healthcare and for the Takeda, Roche, and Seagen Pharmaceutical Companies, and has received institutional research support from Siemens Healthcare. B.P.L. and J.B.A. receive royalties from Elsevier, Inc. as an associate academic textbook editor and author. S.D. is a consultant of Doai and received research support from Tplus and Medibloc. M.K.K. has received institutional research support from Siemens Healthineers, Coreline Inc., and Riverain Tech Inc. J.M.C. was partially supported by a grant from the Korea Health Technology R&D Project through the Korea Health Industry Development Institute (KHIDI) funded by the Ministry of Health & Welfare, Republic of Korea (HI19C1057). The remaining authors declare no competing interests.
