## [Peer Review File · Nature Communications]

Reviewers' Comments:

Reviewer #1:

Remarks to the Author:

The authors propose to address the challenges around large scale dataset annotation using an "atlas-based method". They validate this approach on several large open source datasets.

I found the name "atlas-based method" to be somewhat confusing. Frequently "atlas" based labelling refers to the use of a pre-defined anatomical atlas with labelling of novel instances being achieved by co-registration of the atlas to an image (this is common in neuro-imaging). Consider re-wording or further elaborating on the concept.

The writing is a little unclear as to how the xAI model is actually trained (is it semi-supervised? Weakly supervised on the NLP labels?).

It wasn't obvious to me why NLP was being utilized to provide silver standard labels across the training dataset.

The authors frequently use the word "features" to describe their labels (e.g., pneumonia, cardiomegaly, etc...) and it further makes it difficult to follow the manuscript.

Fig. 1a's "mode selection" is also referenced in Supplementary Table 3 and is a key part of the manuscript, but I don't see it discussed anywhere other than a brief mention on Line 335 and 546? This should be explained more.

The methods discuss using a DenseNet-121 with BCE loss and labels from the NLP, but how does this fit into the "atlas-system"?

Reviewer #2:

Remarks to the Author:

MS number: NCOMMS-21-33133

#General Comments

In this study, the authors presented an automated labeling method for Chest X-ray images using the proposed "probability of similarity"(pSim) metric. They applied the method for detection of five different pathological findings to three large open-source datasets and compared the results to those of 7 human expert radiologists. The study is interesting and well-written, but has some drawbacks. The major strengths and weaknesses are listed as follows.

#Strength of this study

1. A method for automated labeling called patch similarity has been proposed, and the distribution of positive and negative samples of the patches created from CAM to calculate it also appears to be significantly differentiated.
2. Achieved good performance in automated labeling in three different open-access Chest X-ray datasets, equalling or exceeding that of human expert radiologists.

#Weakness of this study

1. Since the "patch similarity" is the key method, it is recommended to compare the performance of using patch similarity only, confidence probability only, and both(pSim). There is nothing new in automated labeling using confidence probability only, so the patch similarity should have significant additive value.
2. Since the proposed method strongly depends on the performance of the model trained with institutional data, it should be shown whether it consistently shows meaningful results in various cases (e.g. using 25%, 50%, 75% and 100% of the training dataset, using different model architecture families such as ResNet, EfficientNet, ...).
3. In page 6, line 109, it is said that the number of Chest PAs in the MIMIC dataset is 71,223, but in the metadata file provided by MIMIC, there are more than 90k cases where the "ViewPosition"

column value is "PA". Also, in the CheXpert dataset, there are 29,420 PAs, but in some cases, multiple images from a single study are included. Therefore, it is necessary to describe in detail how the data selection was made in the open-access data set.

4. It should be specified how the data labeled with the proposed automated labeling method will be used. For example, if the goal is to improve the model performance by adding newly labeled data to the training dataset, it is recommended to show whether there is actually any meaningful improvement through an experiment.

5. The performance of the proposed "patch similarity" depends on CAM, so in page 27, line 480-484, processes such as removing noise components of a single CAM are introduced to improve the localization performance. Recently, there are some methods developed for weakly supervised semantic segmentation using CAM. It is recommended to implement and compare these methods to clarify whether the improvement of the localization performance is related with the performance of the proposed method.

- IRNet

Ahn, Jiwoon, Sunghyun Cho, and Suha Kwak. "Weakly supervised learning of instance segmentation with inter-pixel relations." Proceedings of the IEEE/CVF Conference on Computer Vision and Pattern Recognition. 2019.

- Puzzle-CAM

Jo, Sanghyun, and In-Jae Yu. "Puzzle-CAM: Improved localization via matching partial and full features." arXiv preprint arXiv:2101.11253 (2021).

6. There are both PA and AP in the institutional dataset for development of the model, but only PAs in the open-access dataset is used.

7. Since the proposed method seems nothing more than a classification model and its external validation on open-access datasets, description to clarify which aspects make it to be called automated labeling method rather than just a classification method is recommended.

#Minor

1. Page 39, Supplementary Figure 1 | CXR data collection for the XAI model development : It is recommended to maintain consistency using either xAI or XAI.

Reviewer #3:

Remarks to the Author:

The authors proposed a quantitative explainable atlas-based AI model for the automated labeling of chest X-ray images. Firstly, they trained an explainable AI model (xAI) on a large-scale institution dataset by the automated NLP assignments. Then, they applied this xAI model to label five chest X-ray imaging features on three open-source datasets. To achieve this, they introduced a "probability-of-similarity" (pSim) metric that is calculated by comparing each feature to its training-set derived reference atlas. Based on this metric, they demonstrated that the proposed xAI model could automatically label the external datasets to a user-selected, arbitrarily high level of accuracy, equaling or exceeding that of human experts. Based on the presentation of the manuscript, I have the following comments:

Major comments:

1. In Fig.2, the pSim threshold for each feature is chosen by the lowest values that can achieve 100% PPV and NPV. However, the statistical results in Fig.2 were derived from a small sample batch (n=100). Besides, the results in Table 1 are based on the optimal pSim in this small batch. It is necessary to evaluate its generalization ability to large-scale samples.

2. The pSim value is very sensitive to the different features, varying from 0.15 to 0.65. Could the authors give more suggestions on how to set this pSim to another chest x-ray imaging feature?

3. As the authors said, the proposed xAI model is limited to its initial training set, i.e., NLP assignments. There is no doubt that the noise label in the initial training set would have a bad effect on the whole system. Could the proposed automated labeling via pSim be re-used to refine the initial training set (remove noise)?

4. There are several large-scale open-source CXR datasets with clean labels, like NIH. Based on these datasets, a strong xAI model could also be well-trained. Why the authors collected another institution's dataset? What's the difference or advantages of the in-house dataset? Will it be released for research purposes? Is the performance of xAI model trained on the in-house dataset

with noise labels better than that trained on an open-source dataset like NIH with clean labels?

5. It is necessary to perform the ablation study on the model designs, like the weights for target view position in the weighted loss, removing noise components of CAM, and so on.
6. The test set with only 2000 cases (1000 AP and 1000 PA) is insufficient for model evaluation. Could the authors evaluate the xAI model on three open-source datasets with ground truth?

Minor comments:

1. The scripts in the figures are fuzzy. It would be better to save figures as the vector diagram.

REVIEWER COMMENTS

Reviewer #1 (Remarks to the Author):

The authors propose to address the challenges around large scale dataset annotation using an “atlas-based method”. They validate this approach on several large open-source datasets.

Comment 1 I found the name “atlas-based method” to be somewhat confusing. Frequently “atlas” based labelling refers to the use of a pre-defined anatomical atlas with labelling of novel instances being achieved by co-registration of the atlas to an image (this is common in neuroimaging). Consider re-wording or further elaborating on the concept.

Response 1 Thank you for this useful question. We have clarified our use of the term “atlas-based” by replacing this phrase with “model-derived atlas-based method”, to avoid confusion regarding the use of pre-defined anatomical atlases, as requested. We have updated this throughout the manuscript (. We have also cited previous work that used a similar definition to put this in context (reference #1).

Comment 2 The writing is a little unclear as to how the xAI model is actually trained (is it semi-supervised? Weakly supervised on the NLP labels?).

Response 2 Thank you; we have clarified that model training is neither weakly supervised nor semi-supervised, but rather “supervised”, at two location in the manuscript: (1) lines 118 - 122, “Our xAI model was trained by supervised learning with a total training dataset of 138,686 CXRs and achieved a mean Area Under the Receiver Operating Characteristic (AUROC) curve [10] of 0.95 ± 0.02 for detection of the five clinical output labels (Supplementary Table 2) in our initial, independent test set (Methods)”, and (2) lines 548 - 550, “The pretrained model, available from the official repository in Pytorch [38, 39], was fine-tuned by supervised learning with our training dataset and the NLP’s labels”.

Comment 3 It wasn’t obvious to me why NLP was being utilized to provide silver standard labels across the training dataset.

Response 3 Thank you; as explained in the response to comment 2, we used supervised learning with the NLP labels to train our xAI models, which allowed us to leverage our large training dataset of 138,686 CXRs.

Comment 4 The authors frequently use the word “features” to describe their labels (e.g., pneumonia, cardiomegaly, etc...) and it further makes it difficult to follow the manuscript.

Response 4 Thank you for this useful comment. We have clarified our terminology so that we will use the phrase “clinical output labels” to describe the five categories of: Cardiomegaly, Atelectasis, Pulmonary Edema, Pneumonia, and Pleural Effusion.

Comment 5 Fig. 1a’s “mode selection” is also referenced in Supplementary Table 3 and is a key part of the manuscript, but I don’t see it discussed anywhere other than a brief mention on Line 335 and 546? This should be explained more.

Response 5 Thank you for this helpful comment. We have expanded and clarified our discussion of “mode selection” in Figure 1 (line 75), Table 3 (lines 696 – 701), and manuscript (lines 669 – 677). Specifically:

“Additional functionality of our model design includes a “mode selection” algorithm, which, using the selected pSim threshold value, can be used to either: (1) determine the image label (positive, negative, or unlabeled) within a given level-of-confidence if the pSim value for a class is greater than the selected threshold (“self-annotation mode”), or (2) alert the human user if the pSim falls below the selected threshold for level-of-confidence (“re-annotation mode”). Although the “re-annotation mode” was not applied to our current study, this has the potential to be of value in future applications & deployment of our model, as part of its “explainability” functionality (more details regarding pSim “mode selection” are provided in Methods Table 3”).

Table 3 | “Mode selection” for automated labeling method. The “mode selection” algorithm can be used to either: (1) determine the image label (positive, negative, or unlabeled) within a given level-of-confidence if the pSim value for a class is greater than the selected threshold (“self-annotation mode”), or (2) alert the human user if the pSim falls below the selected threshold for level-of-confidence (“re-annotation mode”).

Mode selection algorithm
Input: predicted probability for c-class (y^c), $Confidence_P$, $Confidence_N$, and patch similarity %[step-1] To divide into two groups by y^c and TH_{pos} : positive or negative candidates If $y^c \geq TH_{pos}$: then %[step-2] To decide mode and annotation for the positive candidates % Probability of Similarity, $pSim$ $pSim = 2 \cdot Confidence_P \cdot pSimilarity / (Confidence_P + pSimilarity)$ If $pSim \geq pSim$ threshold value (PPV, NPV=1): then Mode = Self-annotation mode Label = 1 %Positive label Else Mode = Re-annotation mode Label = -1 %unlabeled Else %[step-2] To decide mode and annotation for the negative candidates $pSim = Confidence_N$ If $pSim \geq pSim$ threshold value (PPV, NPV=1): then Mode = Self-annotation mode Label = 0 %Negative label Else

Mode = Re-annotation mode
Annotation = -1 %unlabeled

Comment 6 The methods discuss using a DenseNet-121 with BCE loss and labels from the NLP, but how does this fit into the “atlas-system”?

Response 6 Thank you; we have updated our Figure 1 (lines 78 – 83) and Table 3 captions (lines 696 – 700) to clarify this, as per our response above to Reviewer 1 Comment 5. We have added lines 78 - 83 to the updated Figure 1, as follows:

“a Our quantitative model-derived atlas-based explainable AI system calculates a “probability of similarity” (pSim) value for automated labeling, based on the harmonic mean between the patch similarity and the confidence calculation values. The resulting pSim metric can be applied to a “mode selection” algorithm, to either label the test images to a selected threshold-of-confidence or alert the user that the pSim value falls below this selected threshold.”

Reviewer #2 (Remarks to the Author):

#General Comments

In this study, the authors presented an automated labeling method for Chest X-ray images using the proposed “probability of similarity”(pSim) metric. They applied the method for detection of five different pathological findings to three large open-source datasets and compared the results to those of 7 human expert radiologists. The study is interesting and well-written, but has some drawbacks. The major strengths and weaknesses are listed as follows.

#Strength of this study

1. A method for automated labeling called patch similarity has been proposed, and the distribution of positive and negative samples of the patches created from CAM to calculate it also appears to be significantly differentiated.
2. Achieved good performance in automated labeling in three different open-access Chest X-ray datasets, equalling or exceeding that of human expert radiologists.

#Weakness of this study

Comment 1 Since the “patch similarity” is the key method, it is recommended to compare the performance of using patch similarity only, confidence probability only, and both(pSim). There is nothing new in automated labeling using confidence probability only, so the patch similarity should have significant additive value.

Response 1 Thank you for this excellent question, which we feel has significantly improved our analysis by suggesting added value of using “pSim” over either (1) “patch similarity” (based on CAM calculations, related to “focal” spatial localization) or (2) confidence probability (related to the “global” probability distribution of the final model output labels), alone. Our new analysis supporting this is shown in the new Figure 7 (lines 309 – 327) and manuscript (lines 199 – 213); it is also noteworthy, per Wikipedia, that “since the harmonic mean of a list of numbers tends strongly toward the least element of the list, it tends (compared to the arithmetic mean) to mitigate the impact of large outliers and aggravate the impact of small ones”. Our new Figure 7 (lines 309 – 327) caption is as follows:

Fig. 7 | Performance comparison of “confidence probability”, “patch similarity”, and “pSim” in assigning true-positive model output labels for cardiomegaly, atelectasis, pulmonary edema, pneumonia, and pleural effusion. We compared the “true positive capture rate” (TPCR) performance for each of the five clinical output labels, using confidence probability alone (reflecting the “global” probability distribution of the output labels), patch similarity alone (reflecting the “focal” spatial localization of the output labels), and pSim (reflecting the harmonic mean between the confidence probability and patch similarity, as per Figure 1). These results are noteworthy in that the two model output labels that reflect high inter-rater agreement of imaging findings – cardiomegaly and pleural effusion, as per Figure 6 – show good agreement between the three confidence-level metrics, with high TPCR’s for each. For the two output labels that show lower inter-rater agreement per Figure 6 - pneumonia and pulmonary edema – pSim performance significantly exceeds that of patch similarity for both, and that of confidence probability for pneumonia but not pulmonary edema. This difference is likely attributable to the fact that “patch similarity” is more sensitive for the detection of focal, regional imaging findings (e.g., as seen with the clinical diagnosis of pneumonia), whereas “confidence

probability” is more sensitive for the detection of global findings (e.g., as seen with the clinical diagnosis of pulmonary edema). The results for atelectasis, typically a more “focal” than “global” finding on CXR, may be similarly explained.

Comment 2 Since the proposed method strongly depends on the performance of the model trained with institutional data, it should be shown whether it consistently shows meaningful results in various cases (e.g. using 25%, 50%, 75% and 100% of the training dataset, using different model architecture families such as ResNet, EfficientNet, ...).

Response 2 Thank you for this important question regarding the relationship between performance consistency, generalizability, dataset size, and architecture. Regarding architecture, we performed a new analysis, shown in Supplementary Figure 2 (lines 869 – 875) and manuscript (lines 215 – 220), that reveals excellent consistency between our current model and three additional, different model architectures, including ResNet-50 [13], MobileNet v2 [14], and MnasNet [15]. Regarding dataset size and heterogeneity, Table 1 and new Table 2 (lines 345 – 356) similarly suggest consistent, robust generalizability (see also response to Reviewer 2 Comment 4). Our caption for new Supplementary Figure 2 (lines 869 – 875) is as follows:

Supplementary Figure 2 | Performance comparison between three additional, different model architectures: RestNet 50, MobileNet v2, and MnasNet. We replicated the analysis shown in Figure 2 by retraining three new models with the following three architectures, ResNet-50 [13], MobileNet v2 [14], and MnasNet [15], using the same training/validation/test datasets, parameters, and environments as for our primary DenseNet-121 model (Methods). Our results show consistent performance, similar to that of Figure 2, for all three new models.

Ensemble performance (AUROC)	DenseNet-121	ResNet-50	MobileNet v2	MnasNet
Category	Testset (Annotation)	Testset (Annotation)	Testset (Annotation)	Testset (Annotation)
Cardiomegaly	0.97	0.97	0.98	0.97
Atelectasis	0.94	0.94	0.94	0.94
Pulmonary edema	0.98	0.97	0.98	0.98
pneumonia	0.90	0.90	0.90	0.91
Pleural effusion	0.98	0.99	0.98	0.98
mean AUROC	0.95	0.95	0.96	0.96

Comment 3 In page 6, line 109, it is said that the number of Chest PAs in the MIMIC dataset is 71,223, but in the metadata file provided by MIMIC, there are more than 90k cases where the “ViewPosition” column value is “PA”. Also, in the CheXpert dataset, there are 29,420 PAs, but in some cases, multiple images from a single study are included. Therefore, it is necessary to describe in detail how the data selection was made in the open-access data set.

Response 3 Thank you for this question, which helps to clarify our data selection process. Although the external datasets contained both AP and PA views, our study was conducted with PA views only, primarily for consistency and convenience, as well as to minimize potential confounding variables. We have added the following manuscript (lines 687 – 694) of the Methods section:

“Although the external datasets contained both AP and PA views, our study was conducted with PA views only, for both consistency/convenience and to minimize potential confounding variables. Specifically, from the CheXpert v1 (n=223,414) and NIH (n=112,120) datasets, which contain ‘PA’ labels in their metadata files, we collected 29,420 and 67,310 PA CXRs respectively. From the MIMIC v1 (n=369,188) dataset, which did not have clear labels, we applied an internal model to distinguish between PA and AP projections, which returned 71,223 PA CXR’s (specificity 0.999, sensitivity = 0.998)”

In addition, please see the response to Reviewer 3, Comment 6.

Comment 4 It should be specified how the data labeled with the proposed automated labeling method will be used. For example, if the goal is to improve the model performance by adding newly labeled data to the training dataset, it is recommended to show whether there is actually any meaningful improvement through an experiment.

Response 4 Thank you for this very important question, which addresses the added value of our approach for using the auto-labeled datasets for fine-tuning. Specifically, new Table 2 (lines 345 – 356) and manuscript (lines 222 – 228) show equal or improved performance – on more generalized datasets than the original model, which was trained on local data only - when comparing the original model ensemble performance (top row Table 2) with the fine-tuned model ensemble performance. Caption for the new Table 2 (lines 345 – 356) is as follows:

Table 2 | Added value of fine-tuning with auto-labeled datasets for model generalizability and performance improvement. To demonstrate the ability of our system to generalize to external datasets at a user selected level of performance, we used the auto-labeled exams selected from the three public datasets shown in Table 1 for fine-tuning of our model. The performance of the six DenseNet-121 models used to create the ensemble of the original model are shown in the top row of Table 2, for each of the five clinical model outputs. The results after fine-tuning using the same environments and hyper-parameters (learning rate 10^{-8} , n=31,020 CXR’s with at least one positive label), are shown in the bottom row of Table 2. Performance for each of the model outputs was preserved or improved on the more generalized model after fine-tuning using the auto-labeled exams.

Original Model	Model 1	Model 2	Model 3	Model 4	Model 5	Model 6	Ensemble
Cardiomegaly	0.972	0.983	0.976	0.998	0.997	1.000	0.994
Atelectasis	0.932	0.946	0.936	0.949	0.928	0.957	0.954
Pulmonary edema	0.949	0.930	0.933	0.968	0.956	0.967	0.960
pneumonia	0.905	0.818	0.869	0.925	0.884	0.912	0.908
Pleural effusion	0.993	0.985	0.984	0.997	0.996	0.996	0.998
AUROC	0.950	0.932	0.940	0.967	0.952	0.966	0.963

Finetuned Model	Model 1	Model 2	Model 3	Model 4	Model 5	Model 6	Ensemble
Cardiomegaly	0.972	0.986	0.988	0.997	0.998	0.998	0.998
Atelectasis	0.942	0.974	0.948	0.960	0.946	0.963	0.965
Pulmonary edema	0.977	0.919	0.954	0.967	0.964	0.971	0.968
pneumonia	0.923	0.870	0.892	0.933	0.894	0.918	0.930
Pleural effusion	0.994	0.994	0.983	0.998	0.998	0.998	0.998
average score	0.961	0.949	0.953	0.971	0.960	0.970	0.972

Comment 5 The performance of the proposed “patch similarity” depends on CAM, so in page 27, line 480-484, processes such as removing noise components of a single CAM are introduced to improve the localization performance. Recently, there are some methods developed for weakly supervised semantic segmentation using CAM. It is recommended to implement and compare these methods to clarify whether the improvement of the localization performance is related with the performance of the proposed method.

Response 5 Thank you for this interesting and intriguing suggestion, which we have already partly addressed in our response to Reviewer 2 Comment 1, where we created a new Figure 7 (lines 309 – 327) that compares “pSim” to both “patch similarity” (focal, CAM related) and “confidence probability” (global related). The use of CAM segmentation in our approach to auto-labeling is specifically intended to contribute to the calculation of “pSim”; indeed, before selecting ensemble CAM at a threshold of 0.5 for this purpose, we considered other methods for focal localization, including but not limited to: GradCAM, Guided GradCAM, Saliency Map, and DeepLIFT, as shown in the example below:

Additionally, we have expanded on the Reviewer 2 Comment 1 response, by adding the following two supplementary references (lines 622 – 623 and 823 – 827):

[42] Ahn, J., Cho, S. and Kwak, S. Weakly supervised learning of instance segmentation with inter-pixel relations. In *Proceedings of the IEEE/CVF Conference on Computer Vision and Pattern Recognition*, 2209-2218, (2019).

[43] Jo, S. and Yu, I.J. Puzzle-CAM: Improved localization via matching partial and full features. Preprint at <https://arxiv.org/abs/2101.11253> (2021).

Moreover, we have also created a new supplementary Figure 3 (lines 876 – 883), in which we show representative examples of how different CAM ensemble method thresholds impact the attention map visualization of the 5 different clinical output labels. Specifically, the caption for new Supplementary Figure 3 (lines 876 – 883) is as follows:

Supplementary Figure 3 | Representative impact of varying ensemble CAM method thresholds on attention map visualization. We considered three thresholds (0.3, 0.5, and 0.7, columns 2-4) for the ensemble CAM method (**Methods**), compared to a single CAM method without noise reduction (column 1), for each of the 5 different clinical output labels. Visual review of several such examples from each category by three human experts (**Methods**) suggested that an ensemble CAM threshold of 0.5 optimally correlated with the reference standard pathology present on the CXR source images.

Comment 6 There are both PA and AP in the institutional dataset for development of the model, but only PAs in the open-access dataset is used.

Response 6 Thank you for this important question. Please refer to our response to Reviewer 2 Comment 3, which discusses this issue. Moreover, we have created a new Supplementary Table 3 (lines 856 – 863), which compares model performance when training using various combinations of PA and AP CXR's for the weighted loss function. The AUROC performance showed minimal differences between using PA CXR's only and three different weightings. This result is in keeping with our clinical experience of CXR interpretation by human experts. The new Supplementary Table 3 (lines 856 – 863) caption is as follows:

Supplementary Table 3 | Comparison of model performance when training using different combinations of PA and AP CXR's. To assess the impact on model performance by training using PA CXR's only, we compared performance when training using various combinations of PA and AP CXR's for the weighted loss function ($W = 1.1, 1.5, \text{ and } 2.0$). The AUROC performance showed minimal differences between using PA CXR's only and three different weightings. This result is in keeping with our clinical experience of CXR interpretation by human experts.

AUROC (Valid Dataset)	Only PA CXRs	PA + AP CXRs		
Hyper-parameter	-	W=1.1	W=1.5	W=2.0
Fracture	0.778	0.790	0.810	0.806
Non-fracture	0.855	0.849	0.856	0.865
Diaphragm	0.897	0.897	0.888	0.899
Foreign body	0.888	0.892	0.895	0.894
Aorta	0.884	0.883	0.881	0.880
Cardiomegaly	0.922	0.920	0.919	0.921
Hilar area	0.883	0.924	0.895	0.927
Mediastinum	0.927	0.944	0.927	0.930
Cavity/Cyst	0.814	0.853	0.854	0.845
Emphysema	0.921	0.921	0.919	0.929
Atelectasis	0.861	0.868	0.866	0.864
Nodule/mass	0.715	0.718	0.720	0.714
Other interstitial opacity	0.793	0.803	0.798	0.798
Pulmonary edema	0.932	0.933	0.932	0.933
pneumonia	0.789	0.802	0.800	0.800
Decreased lung volume	0.882	0.881	0.883	0.883
Increased lung volume	0.896	0.891	0.885	0.892
Other pleural lesions	0.847	0.852	0.848	0.858
Pleural effusion	0.965	0.967	0.964	0.966
Pneumothorax	0.882	0.884	0.890	0.895
mean AUROC	0.867	0.874	0.871	0.875

Comment 7 Since the proposed method seems nothing more than a classification model and its external validation on open-access datasets, description to clarify which aspects make it to be called automated labeling method rather than just a classification method is recommended.

Response 7 Thank you for this very important question, which gives us the opportunity to both clarify and underscore what distinguishes our approach to auto-labeling from “just another” straightforward classification method.

As noted in our abstract and introduction, approaches to date for “automated” annotation of large datasets, both within institutions and public, have primarily “focused on labor-intensive, manual labeling of subsets of these datasets to be used to train new models”. The accuracy of such an approach can be limited not only by the: (1) baseline performance of the index-model, but also by (2) differences in the case mix and image quality of the external datasets. Moreover – as demonstrated by the results of our study – (3) it cannot be assumed that the labels provided with public databases are accurate or “clean”; for example, in some public datasets, such labels may have been generated from potentially “noisy” NLP derived annotation, without validation by an appropriate “platinum level” reference standard.

Given this, our updated abstract, introduction, and discussion sections have been clarified as follows: “We describe a method for standardized, automated labeling based on *similarity* to a previously validated, explainable AI model (xAI), using a model-derived-atlas based approach, for which the user can specify a *quantitative threshold for a desired level of accuracy, the ‘probability-of-similarity’ (pSim) metric*. We showed that our xAI model, by calculating the pSim values for each clinical output label based on *comparison to its training-set derived reference atlas*, could automatically label the external datasets to a *user-selected, high level of accuracy*, equaling or exceeding that of human experts. We additionally showed that, by *fine-tuning the original model using the automatically labeled exams for retraining, performance could be preserved or improved, resulting in a highly accurate, more generalized model.*”

Furthermore, as suggested by the reviewers, we have demonstrated that our automated labeling approach has the following advantages, which may help distinguish it from standard classification methods:

- (1) The use of a *quantitative “pSim” threshold* may have benefits over either “patch similarity” (CAM related) or “confidence calculation” alone (new Figure 7), especially notable for those clinical diagnosis output labels - pneumonia and pulmonary edema - that have the lowest inter-rater agreement among experts.
- (2) The resulting, potentially more accurately labeled exams, can then be *used to fine-tune a more generalized model*, with equal or greater performance than the index model.
- (3) Moreover, this performance could be achieved by human expert (“platinum level”) annotation of a *relatively small subset* of exams – in our manuscript only 100 cases – which nonetheless has the potential to provide highly accurate labeling of arbitrarily large external databases, of even hundreds-of-thousands exam. Indeed, such platinum level expert annotation might only need to be done once for any given platform at any given institution, facilitating automated continuous fine-tuning and retraining (see also Reviewer 3 Comment 6, reference 1). More specifically, our approach is to *select the pSim threshold based on the highest value that will result in PPV, NPV =*

1, as per Figure 2. For this to be accomplished with human expert annotation of a relatively small number of exams, it is *important that the randomly selected subset of “positive” and “negative” cases be distributed equally in each of ten pSim value ranges (0-0.1, 0.1-0.2, 0.2-0.3, ..., 0.9-1.0), as per our Figure 2 caption (10 exams per pSim range, for a total of 100 exams presented for human expert review).*

- (4) Another feature that distinguishes our approach from that of a simple classification model relates to explainability versus a black box approach. Specifically, the “pSim” metric *provides a “reality check”, with quantitative feedback* that the model is “doing what it’s supposed to” and performing at a predefined level of accuracy (see also Reviewer 3 Comment 6, reference 2).
- (5) In summary, our approach for *Accurate Automated Labeling of Chest X-ray Images using “Quantitative Similarity to an Explainable AI Model-Derived-Atlas”,* addresses the following potential limitations of manual labeling using standard classification methods:
 - a. Limitations based on reliance on the baseline performance of the index-model, which may have been trained using data been generated from potentially “noisy” NLP derived annotation, without validation by an appropriate “platinum level” reference standard.
 - b. Limitations based on variability & heterogeneity in the case mix and image quality of the external datasets being studied; the labels provided with public, open-access databases cannot be assumed to be accurate or “clean”.
 - c. Given “a” and “b” above, labeling of external datasets for the purpose of retraining using standard “black box” classification methods is likely to be more labor-intensive than with our approach, because each different distinct dataset (e.g., CheXpert, NIH, MIMIC) may require a larger number of manual labels in order to ensure that sufficient representative exams have been sampled. As demonstrated however with our approach, *through use of the pSim threshold for explainability to estimate a quantitative probability of similarity* (point “4” above), a high degree of confidence that sufficient exams have been sampled for accurate retraining might be achievable.

We have added portions of the above discussion to the Abstract (lines 23-25), introduction (lines 61 – 73), discussion (lines 364 – 370, 424 – 440, 506 - 513), and conclusion (lines 515 – 524) of our manuscript; thank you again.

#Minor

Comment 1 Page 39, Supplementary Figure 1 | CXR data collection for the XAI model development: It is recommended to maintain consistency using either xAI or XAI.

Response 1 Thank you for this excellent suggestion; we have made this change throughout the manuscript.

Reviewer #3 (Remarks to the Author):

The authors proposed a quantitative explainable atlas-based AI model for the automated labeling of chest X-ray images. Firstly, they trained an explainable AI model (xAI) on a large-scale institution dataset by the automated NLP assignments. Then, they applied this xAI model to label five chest X-ray imaging features on three open-source datasets. To achieve this, they introduced a “probability-of-similarity” (pSim) metric that is calculated by comparing each feature to its training-set derived reference atlas. Based on this metric, they demonstrated that the proposed xAI model could automatically label the external datasets to a user-selected, arbitrarily high level of accuracy, equaling or exceeding that of human experts. Based on the presentation of the manuscript, I have the following comments:

Major comments:

Comment 1 In Fig.2, the pSim threshold for each feature is chosen by the lowest values that can achieve 100% PPV and NPV. However, the statistical results in Fig.2 were derived from a small sample batch (n=100). Besides, the results in Table 1 are based on the optimal pSim in this small batch. It is necessary to evaluate its generalization ability to large-scale samples.

Response 1 Thank you for this important question regarding generalizability. Your question overlaps with several queries raised by Reviewer 2; please see the response to Reviewer 2 Comments 3, 4, 6, and 7, which address this issue. Indeed, as noted in response Reviewer 2 Comment 7, an important highlight of our approach is that labor-intensive, human expert, “platinum level” manual annotation of only a relatively small sample batch (n=100 in our paper), has the potential to result in highly accurate auto-labeling of arbitrary large external databases.

Comment 2 The pSim value is very sensitive to the different features, varying from 0.15 to 0.65. Could the authors give more suggestions on how to set this pSim to another chest x-ray imaging feature?

Response 2 Thank you again. As noted in our response to Reviewer 2 Comment 7, our approach is to perform human expert, “platinum level” manual annotation of a relatively small number of exams from the public dataset and select the pSim threshold based on the highest value that will result in PPV, NPV = 1, as per Figure 2. For this to be accomplished with human expert annotation of a relatively small number of exams, it is important that the randomly selected subset of “positive” and “negative” cases be distributed equally in each of ten pSim value ranges (0-0.1, 0.1-0.2, 0.2-0.3, ..., 0.9-1.0), as per our Figure 2 caption (10 exams per pSim range, for a total of 100 exams presented for human expert review).

As already discussed in response to Reviewer 2 Comment 4 and shown in our new Table 2 (lines 345 – 356), we have shown that “by fine-tuning the original model using the automatically labeled exams for retraining, performance could be preserved or improved, resulting in a highly accurate, more generalized model.”

Comment 3 As the authors said, the proposed xAI model is limited to its initial training set, i.e., NLP assignments. There is no doubt that the noise label in the initial training set would have a bad effect on the whole system. Could the proposed automated labeling via pSim be re-used to refine the initial training set (remove noise)?

Response 3 Thank you for this very important question; the answer is “yes”, as per our responses above to your comments #'s 1 and 2, and Table 1 (lines 333 – 344) and Table 2 (lines 345 – 356).

Comment 4 There are several large-scale open-source CXR datasets with clean labels, like NIH. Based on these datasets, a strong xAI model could also be well-trained. Why the authors collected another institution's dataset? What's the difference or advantages of the in-house dataset? Will it be released for research purposes? Is the performance of xAI model trained on the in-house dataset with noise labels better than that trained on an open-source dataset like NIH with clean labels?

Response 4 Thank you for this question, which provides the opportunity for us to provide further clarification of our purpose and use case, as well as place our results in an appropriate context for end users. Specifically, our approach has the potential to provide standardized, automatic labels of external datasets, based on their similarity to a previously validated, explainable AI model-derived-atlas, for which the user can specify a quantitative threshold for a desired level of accuracy. What distinguishes our approach, as shown in Figure 3 of our results, is that the labels provided with large-scale open-source datasets cannot be assumed to be “clean” (as per blue-square pooled datapoints on the AUROC curves of Figure 3). Please also see our data and code availability statements in the main manuscript.

Comment 5 It is necessary to perform the ablation study on the model designs, like the weights for target view position in the weighted loss, removing noise components of CAM, and so on.

Response 5 Thank you for this important suggestion; we have performed these analyses as per our responses to Reviewer 2 Comments 1, 2, 5, and 6. In response to these and your queries, we have added new Figures 7 (lines 309 – 327, manuscript lines 199 – 213), Supplementary Figure 2 (lines 869 – 875, manuscript lines 215 – 220), Supplementary Figure 3 (lines 876 – 883), and Supplementary Table 3 (lines 856 – 863).

Comment 6 The test set with only 2000 cases (1000 AP and 1000 PA) is insufficient for model evaluation. Could the authors evaluate the xAI model on three open-source datasets with ground truth?

Response 6 Thank you again for this comment, which provides us the opportunity to further clarify our approach, purpose, methods, and results. This question overlaps with some of those from Reviewers 1 and 2; specifically, please see our response to Reviewer 1 Comments 2, 3, and 5; Reviewer 2 Comments 4 and 7; and Reviewer 3 Comments 1 and 2. Specifically, our approach has the potential to auto-label an arbitrarily large dataset to a pre-defined level of confidence based on quantitative similarity (“pSim”) to a model-derived-atlas, based on human expert, “platinum level” manual annotation of only a relatively small sample batch from the open-source dataset being studied.

Indeed, motivation for our approach using the pSim metric is further underscored by the following recent papers, which we have cited in our discussion section:

(1) *Rauschecker AM, Gleason TJ, Nedelec P, Duong MT, Weiss DA, Calabrese E, Colby JB, Sugrue LP, Rudie JB, Hess CP. Interinstitutional Portability of a Deep Learning Brain MRI Lesion Segmentation Algorithm. Radiol Artif Intell. 2022; 4(1):e200152.* This paper found that, “for a brain lesion segmentation model trained on a single institution’s data, performance was lower when applied at a second institution;

however, the addition of a small amount (10%) of training data from the second institution allowed the model to achieve its full potential performance level at the second institution”. Our approach may facilitate fine-tuning or retraining to a similar or greater level of performance, using considerably less data than 10% of the initial training set, which ultimately has the potential to contribute to continuous learning, as well generalization of models to different scanners and platforms (see also Reviewer 2 Comment 7).

(2) Arun N, Gaw N, Singh P, Chang K, Aggarwal M, Chen B, Hoebel K, Gupta S, Patel J, Gidwani M, Adebayo J, Li MD, Kalpathy-Cramer J. *Assessing the Trustworthiness of Saliency Maps for Localizing Abnormalities in Medical Imaging. Radiol Artif Intell.* 2021 Oct 6;3(6):e200267.

This paper concludes that the “use of saliency maps in the high-risk domain of medical imaging warrants additional scrutiny” and “recommends that detection or segmentation models be used if localization is the desired output of the network”. A noteworthy feature of our approach is its “quantitative” explainability, based on pSim, which (as per Reviewer 2 Comments 1, 2, 5, & 6 and Reviewer 3 Comment 5), may have added value over saliency maps based on “patch similarity” or “confidence calculations” alone.

Minor comments:

Comment 1 The scripts in the figures are fuzzy. It would be better to save figures as the vector diagram.

Response 1 Thank you. We have updated all figures to be in a scalable vector format.

Reviewers' Comments:

Reviewer #1:

Remarks to the Author:

The article has benefited immensely from undergoing a round of review. It is nicely done and I have no further comments.

Reviewer #2:

Remarks to the Author:

Many aspects of the manuscript have been improved, but there are some points that need to be checked.

1. Response 1 and figure 7 seem interesting, and it shows 2 higher, 1 comparable, and 2 lower results of the patch similarity for detection of five labels. It would be better if you can explain the reason why it shows better performance in atelectasis and pneumonia cases and worse in pulmonary edema and pleural effusion than the traditional confidence probability method.

2. On Page 34, Line 638-641, it is written that the patch similarity is calculated using "the mean Euclidean distance of K-nearest patches for the test image". It is obvious that using similarity with test data improves test performance, and using test data in any way in the development stage can be considered unfair. It is recommended to clarify whether the test data used for patch similarity and performance measurements (e.g. results in Table 2) are the same or not. And if it is the same, it is recommended to use a different set.

REVIEWERS' COMMENTS

Reviewer #1 (Remarks to the Author):

The article has benefited immensely from undergoing a round of review. It is nicely done and I have no further comments.

Reply: Thank you again for your useful suggestions.

Reviewer #2 (Remarks to the Author):

Many aspects of the manuscript have been improved, but there are some points that need to be checked.

1. Response 1 and figure 7 seem interesting, and it shows 2 higher, 1 comparable, and 2 lower results of the patch similarity for detection of five labels. It would be better if you can explain the reason why it shows better performance in atelectasis and pneumonia cases and worse in pulmonary edema and pleural effusion than the traditional confidence probability method.

Reply: Thank you for this useful comment which has prompted us to clarify and re-organize the figures, figure legends, and text. Specifically, to more clearly explain the relationship between performance and pSim values for the five different labels, we have re-grouped the labels, throughout the manuscript, according to confidence probability, patch similarity, and pSim results shown in the new Figure 8 (previously Figure 7). This re-grouping consists of cardiomegaly & pleural effusion (new Figure 2, previously Figure 2), pulmonary edema & pneumonia (new Figure 3, previously Figure 2), and atelectasis (new Figure 4, previously Figure 2), with the captions for Figures 2, 3, 4, and 8 updated to reflect the response to this Reviewer 2 / Comment 1 query.

The updated captions explain that *cardiomegaly & pleural effusion* are focal, high-conspicuity regional imaging findings, that therefore not only have a higher true-positive-capture rate (TPCR) performance with patch similarity than with confidence probability (new Figure 8), but that also have very low p-Sim values (corresponding to high levels of reader confidence for detection) based on the 7 expert reader results (Figure 2).

For *pulmonary edema* - the only label for which TPCR performance is better with confidence probability than with patch similarity (Figure 8) - this result is consistent with the fact that confidence probability is more sensitive for the detection of global, non-localized features, which are typical of pulmonary edema findings on CXR (i.e., pulmonary edema is found diffusely throughout the bilateral lung fields). Indeed, regarding expert reader performance, although *pneumonia* is typically more focal than pulmonary edema (and hence has better patch similarity than confidence probability performance in Figure 8), both *pulmonary edema & pneumonia* tend to have more diffuse, low conspicuity, non-specific findings than those of *cardiomegaly & pleural effusion*, reflected by the relatively higher p-Sim values in Figure 3 compared to Figure 2 (corresponding to a more subjective, lower level of reader confidence for these diagnoses versus the more discrete, objective findings of *cardiomegaly & pleural effusion*).

For *atelectasis*, a more discrete, focal, regional CXR finding than *pulmonary edema* or *pneumonia*, both patch similarity and pSim (Figure 8) show good TPCR performance relative to confidence probability.

Both the figure legends and discussion section have been updated with these new clarifications. It is noteworthy that the explanation for the differences in performance between confidence probability, patch similarity, and pSim for the 5 different labels (Figure 8), corresponds so closely with the reader performance and reader variability shown in Figures 2-5 and 7. This not only confirms our “common sense” clinical insight that cardiomegaly & pleural effusion (as well as atelectasis) are high conspicuity objective CXR findings, whereas pulmonary edema & pneumonia are more non-specific subjective assessments, but also underscores the explainability of our model (through assignment of appropriate pSim values for each label) in mirroring human performance, likely attributable to the radiologist based ground truth used for model training. [Pages 26-27, Lines 857-884]

2. On Page 34, Line 638-641, it is written that the patch similarity is calculated using “the mean Euclidean distance of K-nearest patches for the test image”. It is obvious that using similarity with test data improves test performance and using test data in any way in the development stage can be considered unfair. It is recommended to clarify whether the test data used for patch similarity and performance measurements (e.g., results in Table 2) are the same or not. And if it is the same, it is recommended to use a different set.

Reply: Thank you for the opportunity to clarify this important point. Regarding the first sentence, that patch similarity is calculated using “the mean Euclidean distance of K-nearest patches for the *test image*”, confusion may have arisen due to our imprecise use of terminology; specifically, our use of the term “test image” was not intended to imply that we used an image from the *test set*, but rather, that we used a “test image” from the *training set*. We apologize for this imprecise terminology, and to avoid confusion have changed “test images” to “input images” throughout the manuscript. To further clarify, as suggested in this query, we have made explicit in the methods and discussion sections that the test data (i.e., the consensus ground truth of the 7 expert radiologists, Figures 2-5) is completely distinct from the training data used for both the original model and the fine-tuned model (Table 1). [Page 11, Lines 465-470]